# Predicting onset of symptomatic Alzheimer's disease with plasma p-tau217 clocks

Kellen K. Petersen [1], Marta Milà-Alomà[2,3], Yan Li [1], Lianlian Du[4,5,6,7], Chengjie Xiong[8,9], Duygu Tosun [2,3], Benjamin Saef[1], Ziad S. Saad[10], Lei Du-Cuny[11], Janaky Coomaraswamy[12], Yulia Mordashova[11], Carrie E. Rubel[13], Emily A. Meyers [14], Leslie M. Shaw[15], Jeffrey L. Dage [16,17], Nicholas J. Ashton[18,19,20], Henrik Zetterberg [18,21,22,23,24,25], Kyle Ferber[13], Gallen Triana-Baltzer[10], Michael Baratta[12], Erin G. Rosenbaugh[26], Carlos Cruchaga [8,27,28,29], Eric McDade [1,8], David M. Holtzman [1,8,29], John C. Morris[1,8], J. Martin Sabandal[26], Randall J. Bateman [1,8,29,30], Anthony W. Bannon[31], William Z. Potter, Suzanne E. Schindler [1,8,29] ✉, Alzheimer's Disease Neuroimaging Initiative (ADNI)* & On behalf of the Foundation for the National Institutes of Health (FNIH) Biomarkers Consortium Plasma Aβ and Phosphorylated Tau as Predictors of Amyloid and Tau Positivity in Alzheimer's Disease Project Team*

Predicting not just if, but also when, cognitively unimpaired individuals are likely to develop onset of Alzheimer's disease (AD) symptoms would be useful to clinical trials and, eventually, clinical practice. Although clock models based on amyloid and tau positron emission tomography have shown promise in predicting the onset of AD symptoms, a model based on plasma biomarkers would be more accessible. Using longitudinal plasma %p-tau217 (the ratio of phosphorylated to non-phosphorylated tau at position 217) from two independent cohorts (n = 258 and n = 345), clock models were used to estimate the age at plasma %p-tau217 positivity. The estimated age at plasma %p-tau217 positivity was associated with the age at onset of AD symptoms (adjusted $R^2$ of 0.337–0.612) with a median absolute error of 3.0–3.7 years. Notably, the time from %p-tau217 positivity to onset of AD symptoms was markedly shorter in older individuals. Similar models were constructed with data from one p-tau217/Aβ42 immunoassay and four plasma p-tau217 immunoassays. These findings suggest that the time until onset of AD symptoms can be estimated using a single blood test within a margin of error that is acceptable for use in clinical trials.

AD is the most common cause of dementia and is characterized by amyloid plaques primarily comprised of amyloid-β42 (Aβ42) and neurofibrillary tangles comprising tau[1]. The burden of amyloid plaques as quantified by amyloid positron emission tomography (PET) increases for about 10–20 years during the preclinical phase of AD when patients are cognitively unimpaired and then plateaus around the time of symptom onset[2,3]. By contrast, neurofibrillary tangles as measured by tau PET develop later and increase with symptom severity[4,5]. Although treatments for early

symptomatic AD are now clinically available in some countries[6,7], treating patients earlier during the preclinical phase of disease before major neurodegeneration has occurred may be more efficacious[8,9]. Although biomarkers can accurately identify individuals with AD brain pathology, predicting which individuals are likely to develop symptoms is more challenging, and novel predictive modeling approaches are needed.

Interestingly, once amyloid plaques and neurofibrillary tangles start to accumulate, the burden of pathology follows a remarkably

consistent trajectory across individuals[2,10–13]. These consistent trajectories enable the creation of clock models that relate levels of amyloid or tau PET signal to time and allow for estimation of when individuals developed amyloid or tau PET abnormality[2,11–15]. Unlike general biological aging clocks or categorical staging based on multiple biomarkers, clock models track disease progression with a specific biomarker, thereby providing granular and intuitive time-based staging[1,10,13,16]. Clock models allow alignment of trajectories to a reference point (for example, amyloid or tau PET positivity) that reduces heterogeneity compared to models using chronological age alone[13,16]. Furthermore, the age at amyloid or tau abnormality or positivity estimated by clock models can be used to estimate the age at AD symptom onset[10,12,13,15,17]. Knowing not just if, but also when, AD symptoms are likely to manifest would be very useful to clinical trials that aim to prevent or slow progression to symptomatic AD[18].

Amyloid and tau PET are extremely useful tools in research and clinical trials but are expensive and burdensome methods with limited availability[19]. Recently developed blood-based biomarkers are much more accessible and less expensive[20]. The plasma ratio of amyloid-β peptide 42 to 40 (Aβ42/40) decreases very early during the preclinical phase of AD and then plateaus at moderate levels of amyloid plaque burden, resulting in poor correlations with AD symptoms[13,16,21–23]. However, plasma levels of tau phosphorylated at position 217 (p-tau217) and the ratio of phosphorylated to non-phosphorylated tau at position 217 (referred to as %p-tau217) increase throughout the preclinical and early symptomatic phases of AD[13,16,22]. Importantly, plasma measures of p-tau217 and %p-tau217 have high associations not only with amyloid PET[22,24–28] but also with tau PET[24,25,29], brain volumes[23] and cognition[23,28]. Although the concentration of plasma p-tau217 was previously demonstrated to predict risk for cognitive decline and progression to AD dementia[30,31], no published studies have used plasma p-tau217 or %p-tau217 to estimate when individuals will develop onset of AD symptoms.

In the present study, we aimed to use measurements from a single plasma sample to estimate not only the probability of a cognitively unimpaired individual with positive AD biomarkers developing AD symptoms but also when they would be likely to develop symptoms. Primary analyses were performed with the C2N Diagnostics plasma %p-tau217 measure because large longitudinal datasets were available from two cohorts: the Knight Alzheimer's Disease Research Center (Knight ADRC) and the Alzheimer's Disease Neuroimaging Initiative (ADNI). Plasma %p-tau217 has high accuracy in classifying amyloid PET, tau PET and cognitive status; this assay is a component of C2N Diagnostics' clinically available PrecivityAD2 test and is used by some clinical trials[23,25–27,32]. Using longitudinal plasma %p-tau217 data, we created clock models that related %p-tau217 values to time and enabled estimation of the age at %p-tau217 positivity for each individual. Two approaches were used to create %p-tau217 clock models: Temporal Integration of Rate Accumulation (TIRA)[10,13], which integrates the inverse of the modeled rate of change, and Sampled Iterative Local Approximation (SILA)[12,17], which uses discrete rate sampling and Euler's method for numerical integration. Multiple statistical models were used to examine the relationships between the estimated age at %p-tau217 positivity and the onset of AD symptoms. Secondary analyses to assess the generalizability of this approach across plasma biomarkers were performed in the ADNI cohort with the Fujirebio Lumipulse p-tau217/Aβ42 measure that was recently cleared by the US Food and Drug Administration and with four commercially available p-tau217 assays: C2N Diagnostics, Janssen LucentAD Quanterix, ALZpath Quanterix and Fujirebio Lumipulse[23].

## Results

### Development and validation of plasma %p-tau217 clock models
Data from participants with longitudinal plasma %p-tau217 measurements collected at least 1 year apart were considered for potential inclusion in developing the clock models (Supplementary Table 1). When the baseline plasma %p-tau217 sample was collected, 506 individuals from the Knight ADRC cohort had a median age of 67.7 years (interquartile range (IQR) 61.7–72.4 years); 54.2% were female, 35.8% were APOE ε4 carriers and 8.5% were cognitively impaired (defined by Clinical Dementia Rating (CDR) > 0). Knight ADRC participants had a median age of 7.1 years (IQR 5.0–11.0 years) from the first to last plasma collection. Of these participants, 310 of 506 (61.3%) provided three or more plasma samples. The ADNI cohort consisted of 406 individuals with a median age of 72.7 years (IQR 67.9–78.0 years); 49.3% were female, 34.2% were APOE ε4 carriers and 48.3% were cognitively impaired. ADNI participants had a median age of 5.0 years (IQR 4.0-6.5 years) from the first to last plasma collection. Of these participants, 404 of 406 (99.5%) provided three or more plasma samples.

The rate of change in plasma %p-tau217 as a function of the estimated %p-tau217 value at the midpoint of collected samples for each participant was modeled with generalized additive models (GAMs) separately in the two cohorts (Fig. 1a,b). As was described for similar models using amyloid and tau PET[13], intervals with relatively low variance in the rate of change (that is, below the 90th percentile) were identified. These intervals were between plasma %p-tau217 values of 0.29% and 10.45% for the Knight ADRC cohort and between 1.06% and 10.59% for the ADNI cohort (Extended Data Fig. 1a). The intersection yielded an interval of consistent change for plasma %p-tau217 of 1.06% to 10.45%.

For the development of clock models, the cohort was restricted to individuals with longitudinal data within the interval of consistent change (Extended Data Table 1). The Knight ADRC clock cohort included 258 individuals with a median interval between first and last plasma %p-tau217 values of 6.5 years (IQR 3.9–9.8 years). The ADNI clock cohort included 345 individuals with a median interval between first and last plasma %p-tau217 values of 4.5 years (IQR 4.0–6.3 years). Plasma %p-tau217 positivity was defined as more than 4.06% to align with an amyloid PET Centiloid value of 20 (ref. 23). Two methods that have previously been used to create clock models with longitudinal amyloid and tau PET data were used to estimate the years from plasma %p-tau217 positivity: TIRA[10,13] and SILA[12,17]. The clock models relating time to plasma %p-tau217 levels are shown for the Knight ADRC (Fig. 1c) and ADNI (Fig. 1d) cohorts; numerical values are provided in Extended Data Table 2 and can also be accessed via a web-based application: https://amyloid.shinyapps.io/plasma_ptau217_time/#.

Similar clock models were constructed using all longitudinal data (Supplementary Fig. 1). At very low %p-tau217 values (<1.06%), there were highly unstable TIRA time estimates in the ADNI cohort. Longitudinal data from individuals with high %p-tau217 values were sparse (Fig. 1a,b), providing less certain clock estimates for higher values. Furthermore, there were rapid increases in %p-tau217 at high values (>10.45%) that could make time estimates unstable. These features provided further rationale to use clocks constructed with the restricted range of %p-tau217 values (1.06–10.45%).

Plasma %p-tau217 trajectories as a function of age are shown for individuals included in the clock models (Fig. 2a,d) and demonstrate considerable heterogeneity, reflecting that AD pathology begins at widely varying ages. The plasma %p-tau217 trajectories were then re-plotted by years from estimated %p-tau217 positivity (age at plasma collection minus the individual's estimated age at %p-tau217 positivity) for each cohort and clock model (Fig. 2b,c,e,f). The trajectories then demonstrated increased alignment, suggesting relatively consistent patterns of change in plasma %p-tau217 across individuals (Supplementary Video 1).

To assess model performance, the estimated age at plasma %p-tau217 positivity based on the clock models was compared to the age at observed %p-tau217 conversion, which was the average of the age at last negative and first positive %p-tau217 values. For TIRA models, the adjusted $R^2$ was 0.733 in the Knight ADRC cohort and 0.815 in the ADNI cohort; for SILA models, the adjusted $R^2$ was 0.506 in the Knight ADRC

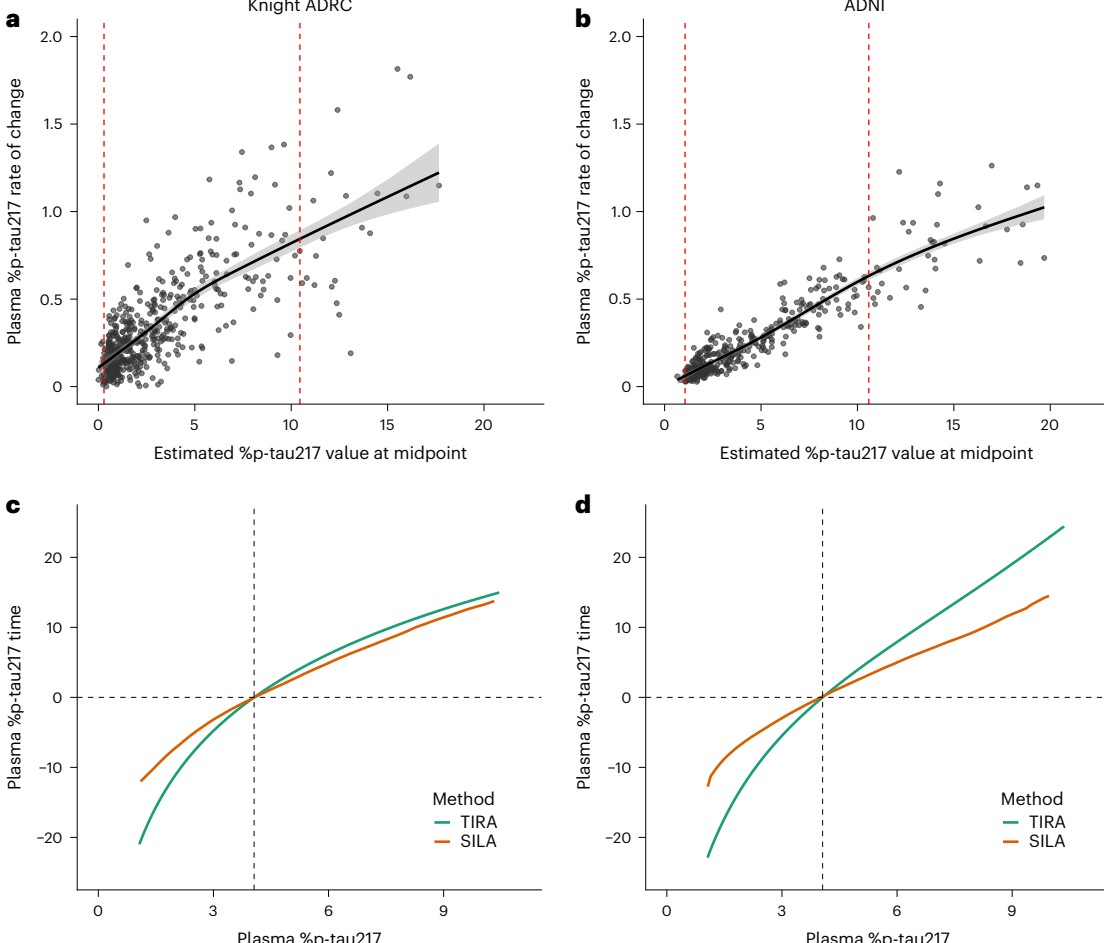

**Fig. 1 | Creation of plasma %p-tau217 clock models.** For the Knight ADRC (**a**) and ADNI (**b**) cohorts, each gray point represents the rate of change in plasma %p-tau217 for an individual as a function of the estimated %p-tau217 value at the midpoint of follow-up. Black solid lines represent GAMs fitting the individual data points, showing the predicted mean rate of change. Gray shading indicates 95% confidence intervals around the fitted GAM curves. Red dashed vertical lines represent the estimated %p-tau217 value at which variance in the rate of change was greater than the 90th percentile of variance in rates of change for the cohort, which identified a range of 1.06–10.45% over which %p-tau217 had relatively consistent change. Clock models relating time to %p-tau217 are shown for the Knight ADRC (**c**) and ADNI (**d**) cohorts and were created using two approaches: TIRA (green) and SILA (orange). The vertical black dashed line represents the plasma %p-tau217 threshold of 4.06%, which aligns with an amyloid PET Centiloid value of 20. The horizontal black dashed line represents the estimated time that an individual has a %p-tau217 value of 4.06%.

cohort and 0.801 in the ADNI cohort (Extended Data Fig. 2). Additionally, we performed cross-cohort comparison of the models and found that the estimated ages at plasma %p-tau217 positivity based on TIRA models implemented on either the Knight ADRC or ADNI cohorts were highly correlated (adjusted $R^2$ of 0.978), and ages based on SILA models from the two cohorts were almost perfectly aligned (adjusted $R^2$ of 0.999) (Supplementary Fig. 2). The estimated ages at plasma %p-tau217 positivity based on the TIRA or SILA models were also highly correlated in the Knight ADRC cohort (adjusted $R^2$ of 0.942) but were less correlated in the ADNI cohort (adjusted $R^2$ of 0.863) (Supplementary Fig. 3). Overall, the clock models estimated similar ages at plasma %p-tau217 positivity regardless of the method or cohort and were consistent with observed conversion ages from %p-tau217 negative to positive.

**Modeling the probability of developing symptomatic AD**

Cox proportional hazards models were used in the longitudinally assessed Knight ADRC cohort to estimate the probability of initially cognitively unimpaired individuals developing symptomatic AD as a function of age (Fig. 3a). Symptomatic AD was defined to align with the clinical diagnosis of symptomatic AD: cognitive impairment (CDR > 0) with an AD syndrome (clinical features consistent with cognitive impairment caused by AD, including amnestic, logopenic aphasia, posterior

cortical dysfunction or dysexecutive presentations[1,33–35]) in the context of biomarkers, indicating the presence of AD pathology[1,35]. The onset of AD symptoms was defined as the first clinical assessment when initially cognitively unimpaired (CDR = 0) individuals with positive AD biomarkers (based on estimated %p-tau217) were found to be cognitively impaired (CDR > 0) with an AD syndrome. Furthermore, AD symptom onset was applied only to individuals who were cognitively impaired (CDR > 0) with an AD syndrome at their last assessment—that is, if an individual had transient cognitive impairment but returned to cognitively unimpaired or had a non-AD diagnosis at their last assessment, the earlier impairment was not considered to be the onset of AD symptoms. To account for variable time between clinical assessments, the time of AD symptom onset was interval-censored between the last cognitively unimpaired assessment and the first symptomatic AD assessment.

Notably, the age at plasma %p-tau217 positivity dramatically affected the time until AD symptom onset. For example, with TIRA-based estimates, participants who became plasma %p-tau217 positive at age 60 had a median time until symptom onset of 20.5 years, whereas participants who became positive at age 80 had a median time until symptom onset of only 11.4 years (Fig. 3b). Cox models for both the Knight ADRC and ADNI cohorts are shown in Supplementary Fig. 4. The probability of symptomatic AD associated with a specific plasma

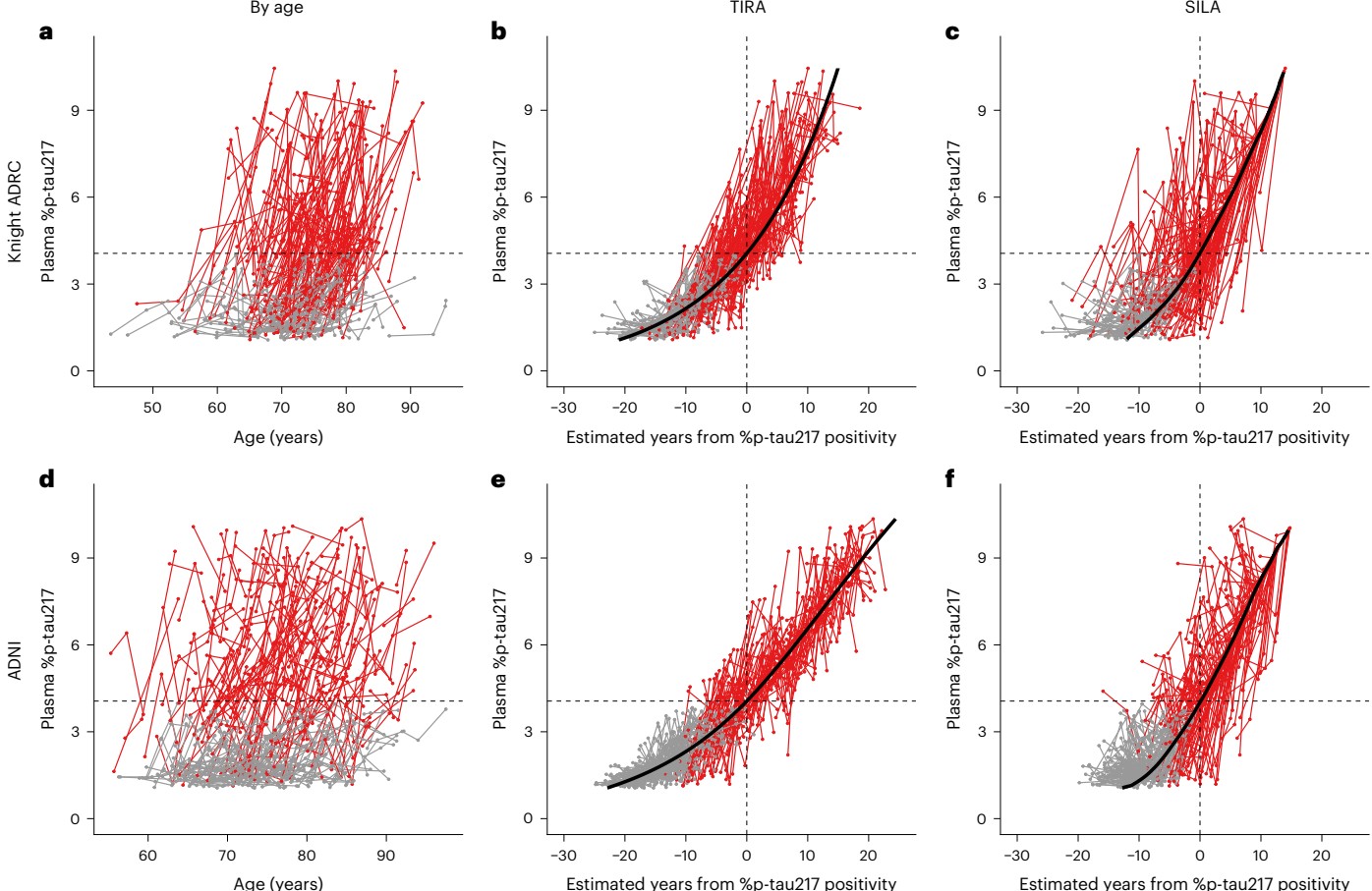

**Fig. 2 | Plasma %p-tau217 trajectories as a function of age and estimated years from %p-tau217 positivity.** Longitudinal plasma %p-tau217 data from the Knight ADRC (**a**–**c**) and ADNI (**d**–**f**) cohorts are shown as a function of age (**a**,**d**) or estimated years from %p-tau217 positivity by TIRA (**b**,**e**) or SILA (**c**,**f**) clock models. Thick black lines represent the clock models shown in Fig. 1c,d; red lines represent individuals with at least one plasma %p-tau217 > 4.06%; and gray lines represent individuals with no plasma %p-tau217 > 4.06%. Horizontal black dashed lines represent the plasma %p-tau217 threshold of 4.06%. Vertical black dashed lines represent the estimated time that an individual had a %p-tau217 value of 4.06%.

%p-tau217 value and age can be accessed via a web-based application: https://amyloid.shinyapps.io/plasma_ptau217_time/.

The discriminative ability of Cox models to rank individuals by their risk of developing symptoms was assessed with a concordance index (C-index), where 0.5 is random prediction, >0.7 is very good prediction, >0.8 is excellent prediction and 1.0 is perfect prediction. For the Knight ADRC cohort, the TIRA-based model yielded a bootstrapped C-index of 0.784 (95% confidence interval: 0.720–0.843), and the SILA-based model had a C-index of 0.790 (95% confidence interval: 0.728–0.847). For the ADNI cohort, the TIRA-based model yielded a C-index of 0.730 (95% confidence interval: 0.622–0.834), and the SILA-based model had a C-index of 0.750 (95% confidence interval: 0.636–0.853). Additional analyses incorporating left-censored participants for those with cognitive impairment at study enrollment confirmed these associations with better discrimination and addressed potential survivor bias (Supplementary Fig. 5). Knight ADRC models achieved bootstrapped C-indexes of 0.821 (95% confidence interval: 0.783–0.857) for TIRA-based models and 0.828 (95% confidence interval: 0.791–0.862) for SILA-based models, whereas ADNI models yielded C-indexes of 0.672 (95% confidence interval: 0.614–0.727) for TIRA-based models and 0.707 (95% confidence interval: 0.651–0.759) for SILA-based models.

**Predicting the age of AD symptom onset**

Next, the estimated age of plasma %p-tau217 positivity was used to model the age at onset of AD symptoms.

For the Knight ADRC cohort, models included 59 individuals using TIRA and 61 individuals using SILA clock models; for the ADNI cohort, models included 20 individuals using TIRA and 22 individuals using SILA (see Supplementary Table 2 for cohort characteristics). The age at onset of AD symptoms was moderately associated with the estimated age at plasma %p-tau217 positivity for both TIRA-based (Knight ADRC adjusted $R^2$ of 0.599; ADNI adjusted $R^2$ of 0.337) and SILA-based (Knight ADRC adjusted $R^2$ of 0.612; ADNI adjusted $R^2$ of 0.470) estimates (Fig. 4 and Supplementary Table 3). Comprehensive model diagnostics confirmed the appropriateness of linear modeling across all cohort–method combinations (Supplementary Table 4). Sensitivity analyses removing outliers demonstrated continued significance of the linear models. The estimated age at AD symptom onset associated with a specific plasma %p-tau217 value and age can be accessed via a web-based application: https://amyloid.shinyapps.io/plasma_ptau217_time/#. Consistent with findings from the Cox models, these models demonstrated that older individuals had a much shorter interval from plasma %p-tau217 positivity to symptom onset (Supplementary Fig. 6).

The effects of *APOE* ε4 carrier status, sex and years of education were also examined, but these variables were either not significant or had minimal effects and, thus, were not included in the models (Supplementary Table 5). Associations between estimated age at plasma %p-tau217 positivity and age at symptom onset were lower when individuals who developed cognitive impairment before %p-tau217 positivity were included, likely because these individuals had non-AD causes of cognitive impairment (Supplementary Fig. 7).

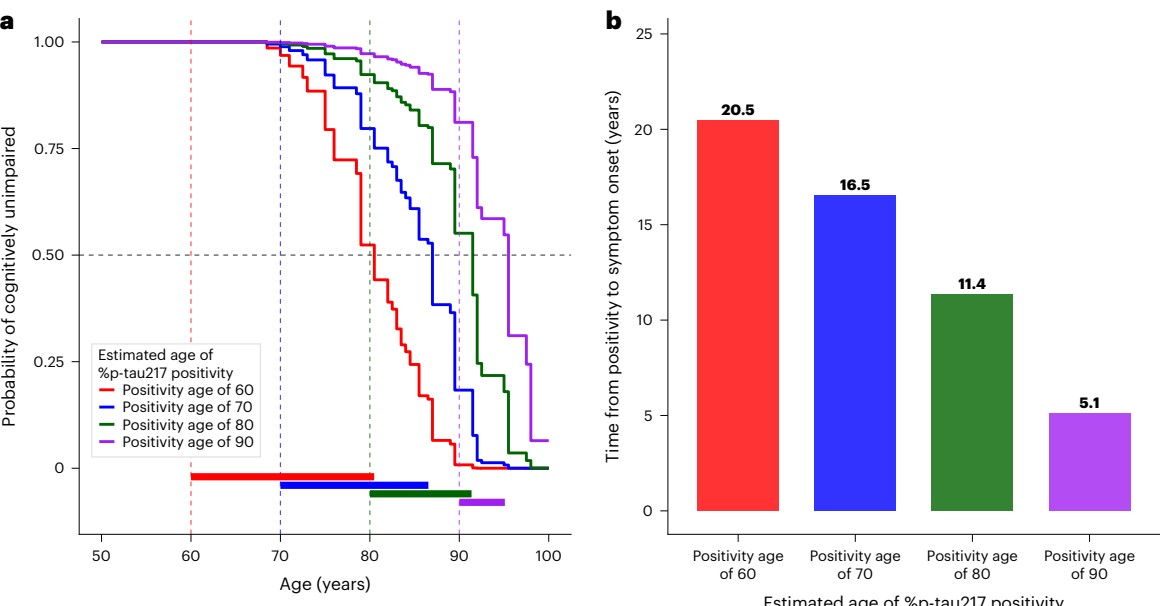

**Fig. 3 | Probability of remaining cognitively unimpaired from AD is related to age at plasma %p-tau217 positivity.** For individuals in the Knight ADRC cohort who were cognitively unimpaired at baseline, a Cox model evaluated the probability of remaining cognitively unimpaired from AD as a function of age, stratified by the age at %p-tau217 positivity based on TIRA (**a**). The estimated time from %p-tau217 positivity until 50% of individuals would be expected to have symptomatic AD is shown as a function of estimated age at %p-tau217 positivity (**b**).

The error in models predicting the age at AD symptom onset based on the estimated age at plasma %p-tau217 positivity was examined (Extended Data Table 3). Within the same cohort, models predicting the age at AD symptom onset had a median absolute error (MdAE) that ranged from 3.0 years to 3.5 years and non-parametric concordance correlation coefficients (CCCs) ranging from 0.771 to 0.839 (CCC > 0.6 is considered good and CCC > 0.8 is considered excellent). For models created in the Knight ADRC cohort and applied to plasma %p-tau217 values in the ADNI cohort, there were moderate associations (adjusted $R^2$ of 0.467 for TIRA and 0.463 for SILA) with an MdAE of 3.0–3.2 years and a CCC of 0.801–0.805. Conversely, for models created in the ADNI cohort and applied to plasma %p-tau217 values in the Knight ADRC cohort, there were also moderate associations (adjusted $R^2$ of 0.509 for TIRA and 0.577 for SILA) with an MdAE of 3.6–3.7 years and a CCC of 0.808–0.820.

### Examining predicted AD symptom onset across longitudinal cognitive assessments

The relationship between predicted AD symptom onset and longitudinal clinical diagnoses was examined. For initially cognitively unimpaired individuals in the Knight ADRC cohort with an estimated age at plasma %p-tau217 positivity by the TIRA-based model (Supplementary Table 6), individuals who became %p-tau217 positive at age 60 were estimated to develop symptomatic AD after 14.0 years, whereas individuals who became %p-tau217 positive at age 80 were estimated to develop symptomatic AD after only 6.2 years (Figs. 4 and 5 Supplementary Table 7). Similar analyses were performed for individuals with an estimated age at plasma %p-tau217 positivity by TIRA or SILA in both the Knight ADRC and ADNI cohorts, including only those who were initially cognitively unimpaired (Supplementary Fig. 8) and all individuals regardless of initial cognitive status (Supplementary Tables 7–9 and Extended Data Fig. 3). Regardless of the clock model used to estimate the age at plasma %p-tau217 positivity, there was a markedly shorter time until symptom onset for individuals who developed %p-tau217 positivity at older ages in both the Knight ADRC and ADNI cohorts.

Progression of initially cognitively unimpaired Knight ADRC participants with a positive plasma %p-tau217 value to cognitive impairment, with either an AD or a non-AD syndrome, demonstrated wide variation in the time until symptom onset (Fig. 6a). Progression of individuals to cognitive impairment as a function of years from estimated plasma %p-tau217 positivity (Fig. 6b) or estimated years from symptom onset (Fig. 6c) is shown for TIRA-based models. Consistent with the other models, binning individuals by estimated age at plasma %p-tau217 positivity demonstrates that older individuals develop symptoms sooner after %p-tau217 positivity (Fig. 6d,e). However, models estimating symptom onset based on age at plasma %p-tau217 adjust for this age effect (Fig. 6f). Similar analyses are shown for SILA-based models in the Knight ADRC dataset (Supplementary Fig. 9) and for TIRA-based and SILA-based models in the ADNI dataset (Supplementary Figs. 10 and 11).

### Alignment of biological AD stages and plasma %p-tau217-derived measures

The alignment of the 2024 Alzheimer's Association biological stages with plasma %p-tau217, years since estimated plasma %p-tau217 positivity and estimated years from symptom onset based on plasma %p-tau217 were examined. Individuals with amyloid PET, tau PET and an estimated age at plasma %p-tau217 were classified according to the biological staging framework: stage A (normal biomarkers), stage B (AD pathologic change), stage C (AD) and stage D (advanced AD). As expected, all three plasma %p-tau217-derived measures increased with advancing biological stage (stage A, stage B, stage C and stage D), demonstrating that %p-tau217 captures the co-evolution of amyloid and tau pathologies across the AD continuum. Estimated years since symptom onset did not as clearly separate the biological stages (Extended Data Fig. 4); it is possible that biological stages may have different relationships with symptoms across age groups.

### Development of additional plasma clocks

To assess whether clock models could be generated using other measures of plasma p-tau217, the same approach was implemented in the ADNI cohort with the Fujirebio Lumipulse p-tau217/Aβ42 measure and with four commercially available p-tau217 assays. First, variance analysis was used to find a range of values where the measure had a relatively consistent rate of change: Fujirebio Lumipulse p-tau217/Aβ42,

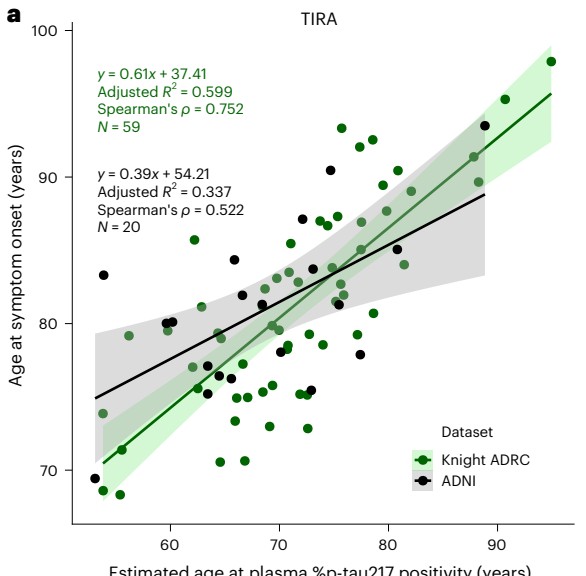

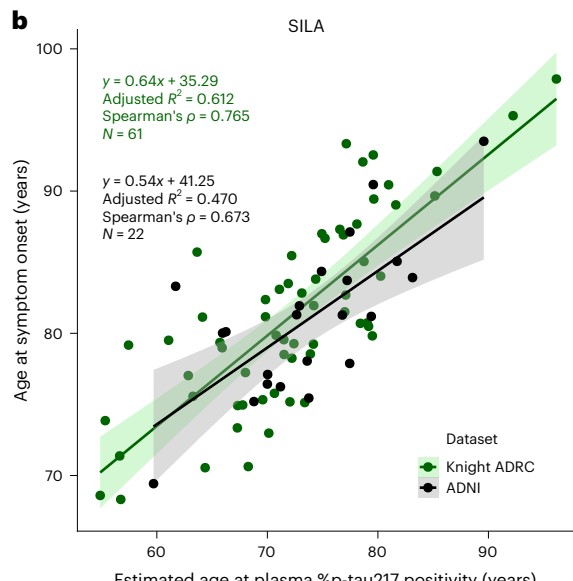

**Fig. 4 | Models for age at symptom onset based on estimated age at plasma %p-tau217 positivity.** Individuals were included who were initially cognitively unimpaired but had a typical AD syndrome at their last assessment and developed symptoms after estimated plasma %p-tau217 positivity. Age at %p-tau217 positivity was estimated using TIRA (**a**) or SILA (**b**) models in the Knight ADRC (green points) or ADNI (black points) cohorts. Each point represents an individual participant. Linear regression lines represent the predicted mean age at symptom onset for each dataset, with shaded bands indicating 95% confidence intervals around the regression lines (green shading for Knight ADRC data and gray shading for ADNI data). Linear regression equations, adjusted $R^2$ values, Spearman's correlation coefficients ($\rho$) and sample sizes ($N$) are shown for each cohort.

0.00−0.02; Janssen LucentAD Quanterix p-tau217, 0.01−0.16 pg ml⁻¹; ALZpath Quanterix p-tau217, 0.10−1.11 pg ml⁻¹; C2N Diagnostics PrecivityAD2 p-tau217, 0.72−7.99 pg ml⁻¹; and Fujirebio Lumipulse p-tau217, 0.02−0.51 pg ml⁻¹ (Extended Data Fig. 1b−e). Previously published positivity thresholds that align with amyloid PET Centiloid 20 were used for the p-tau217 measures[23]; using the same methodology, the threshold for Fujirebio Lumipulse p-tau217/Aβ42 was 0.00631. Next, TIRA and SILA were implemented to create clock models that related time from plasma positivity. Plasma trajectories by age or estimated years from positivity are shown for each plasma measure (Extended Data Figs. 5−7 and Supplementary Figs. 12 and 13). Dynamic visualizations of trajectory alignment for each assay are provided in Supplementary Videos 2−6.

The associations between age at symptom onset and age at plasma positivity were examined: Fujirebio Lumipulse p-tau217/Aβ42, TIRA adjusted $R^2$ of 0.276 and SILA adjusted $R^2$ of 0.584 (Extended Data Fig. 5); Janssen LucentAD Quanterix p-tau217, TIRA adjusted $R^2$ of 0.041 and SILA adjusted $R^2$ of 0.211 (Extended Data Fig. 6); ALZpath Quanterix p-tau217, TIRA adjusted $R^2$ of 0.082 and SILA adjusted $R^2$ of 0.258 (Extended Data Fig. 7); C2N Diagnostics p-tau217, TIRA adjusted $R^2$ of 0.084 and SILA adjusted $R^2$ of 0.301 (Supplementary Fig. 12); and Fujirebio Lumipulse p-tau217, TIRA adjusted $R^2$ of 0.239 and SILA adjusted $R^2$ of 0.450 (Supplementary Fig. 13).

## Discussion

In this study, we demonstrated that clock modeling methods can be used to estimate the age at plasma %p-tau217 positivity and align %p-tau217 trajectories, revealing a relatively consistent change across individuals in %p-tau217 during the preclinical and early symptomatic phase of AD. Furthermore, the estimated age at plasma %p-tau217 positivity was associated with the age at AD symptom onset, enabling prediction of the age at AD symptom onset with a single %p-tau217 value. We found similar results with plasma p-tau217/Aβ42 and p-tau217 measured by immunoassays, demonstrating the generalizability of our approach across different blood biomarker tests.

The estimated years until AD symptom onset based on plasma %p-tau217 had an MdAE of 3−4 years, which would limit its utility for

individual decision-making, but it could still be useful for group-level studies. Clock models could improve selection of participants for clinical trials who are more likely to develop symptoms within the trial period, increasing statistical power and reducing the time needed to demonstrate treatment efficacy. The variance in age at AD symptom onset explained by estimated age at %p-tau217 positivity ranges from 0.337 to 0.599, which is similar to that explained by parental age at onset for autosomal dominant Alzheimer's disease (ADAD) ($R^2 = 0.384$ (ref. 36)). The relatively predictable age at symptom onset has been a major advantage of performing clinical trials in ADAD cohorts[3,37].

It is possible that clock models incorporating plasma %p-tau217 or p-tau217 with other biomarkers, such as eMTBR-tau243 (ref. 38) or biomarkers of cerebrovascular disease[39], may enable greater precision in estimating time until AD symptom onset. Future investigations could also explore continuous cognitive measures that identify subtle cognitive changes that occur before the threshold for clinical diagnosis. More precise models may decrease the error in the estimated years until AD symptom onset to a level that it becomes relevant for individual decision-making, which could have considerable clinical and ethical implications[40]. AD biomarker testing of cognitively unimpaired individuals is currently not recommended outside of research studies or clinical trials due to uncertain benefits and potential risks[19,41–43], and we discourage individuals from using these models to determine their personal estimated age at AD symptom onset.

Notably, we found that older individuals have a markedly shorter time until AD symptom onset after developing plasma %p-tau217 positivity. This is consistent with our previous work predicting symptom onset with amyloid PET[10], where we found that older individuals developed symptoms at a lower amyloid PET burden. Age-related brain changes, including age-related increases in the prevalence of co-pathologies that affect the relationships between clinical symptoms and AD pathology[44], may underlie this effect. As age increases, co-pathologies become more common and may further complicate the interpretation of %p-tau217 levels in older individuals. This finding has major implications for clinical trials: individuals with the same plasma

%p-tau217 values likely have very different risks of developing cognitive impairment over a 3–5-year period depending on their age. Although statistical models typically include age as a covariate, the relationship between plasma %p-tau217 levels and symptom onset is complex and may not be well captured by linear or nonlinear models, although age-stratified analyses may be helpful. Nonlinear mixed-effects models characterize population-level trajectories with individual random effects, but clock models explicitly convert biomarker levels into individualized estimates that are intuitive (for example, years since biomarker positivity) and may reveal important findings such as the marked effect of age at plasma %p-tau217 positivity on the age at AD symptom onset.

Although our clock models use the single biomarker %p-tau217, its strong associations with amyloid and tau PET effectively integrate the pathological processes of amyloid plaques and neurofibrillary tangles into the models. Notably, %p-tau217 dynamics likely capture the intertwined progression of both amyloid and tau pathology. Furthermore, the shorter interval from %p-tau217 positivity to symptom onset observed in older individuals may, in part, reflect the influence of age-related co-pathologies, such as cerebrovascular disease and other neurodegenerative diseases[44]. Recognizing the impact of these additional pathologies is crucial, as they may modify clinical trajectories beyond the core AD pathology. Future work incorporating complementary biomarkers of amyloid, tau and other pathologies will be important for improving the accuracy and applicability of these models.

The %p-tau217 clock models were created by implementing two different mathematical approaches in the single-site Knight ADRC cohort and the multicenter ADNI cohort. Knight ADRC participants were younger, had a much lower rate of cognitive impairment at baseline and were more likely to be *APOE ε*4 carriers. Despite these differences, the TIRA and SILA models generally were aligned, and neither was clearly superior, indicating the robustness of the clock concept for modeling years until AD symptom onset. Still, there were some differences, such as TIRA estimating longer periods compared to SILA, particularly in the ADNI cohort. These findings suggest that it may be helpful to implement multiple approaches when developing clocks and to rigorously evaluate the model fit. We have shared code for the TIRA method (the code for the SILA method is already publicly available from the Betthauser laboratory at the University of Wisconsin), which will facilitate testing of these modeling approaches in additional cohorts and using other measures. We encourage interested investigators to further refine these and other approaches to improve prediction of AD symptom onset. We also recommend that clinical trialists use this code and data to create models tailored to their specific goals—for example, determining p-tau217 values for specific age groups that identify individuals at high risk of developing AD symptom onset within 3 years.

**Fig. 5 | Clinical diagnosis as a function of estimated age at plasma %p-tau217 positivity and years from %p-tau217 positivity.** For initially cognitively unimpaired individuals in the Knight ADRC cohort, the age at plasma %p-tau217 positivity was estimated using the TIRA clock model. Each row represents the longitudinal clinical diagnoses for one individual by estimated years from %p-tau217 positivity (*x* axis). Individuals are sorted vertically by estimated age at %p-tau217 positivity (*y* axis). The point color denotes the clinical diagnosis: blue represents cognitively unimpaired at the assessment; red (AD syndrome/ biomarker positive) represents cognitively impaired at the assessment and a diagnosis of symptomatic AD at their last assessment with symptoms starting after %p-tau217 positivity; purple (AD syndrome/biomarker negative) represents cognitively impaired at the assessment and a diagnosis of symptomatic AD at their last assessment with symptoms starting before %p-tau217 positivity; and orange (non-AD syndrome) represents cognitively impaired and a non-AD diagnosis at their last assessment. The vertical dashed line at 0 represents the estimated time of %p-tau217 positivity. The brown line represents the estimated relationship between %p-tau217 positivity age and estimated symptom onset based on the Knight ADRC model in Fig. 4a.

Despite these promising findings, our approach has limitations. The clock models can only be used for values over which there is a consistent change in %p-tau217 (values between 1.06% and 10.45%). Values outside this range, such as a %p-tau217 of 12%, cannot be used to estimate years until AD symptom onset, although individuals with very high %p-tau217 values are likely at very high risk for developing symptomatic AD. Similarly, very low %p-tau217 values suggest a low likelihood of developing symptomatic AD for many years, but precise estimates of years until AD symptom onset cannot be made. Our

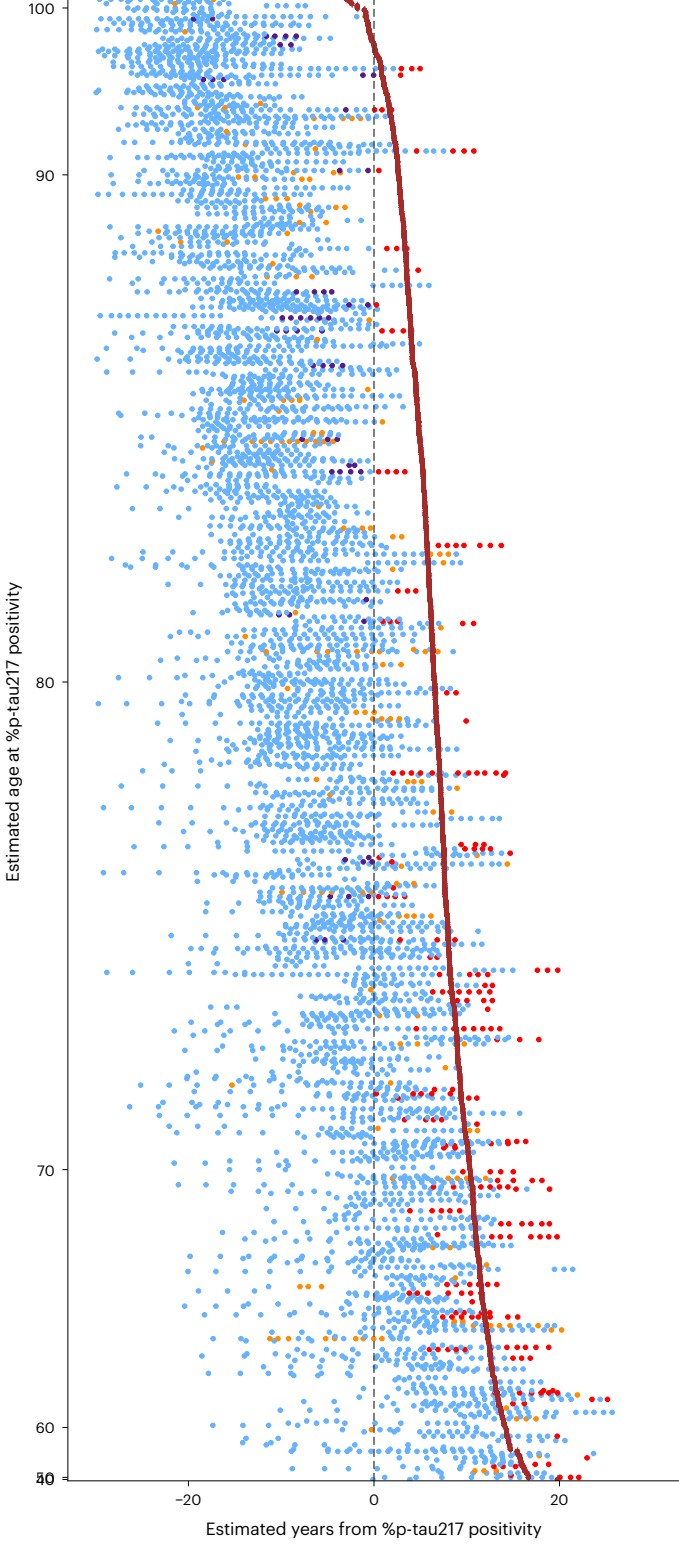

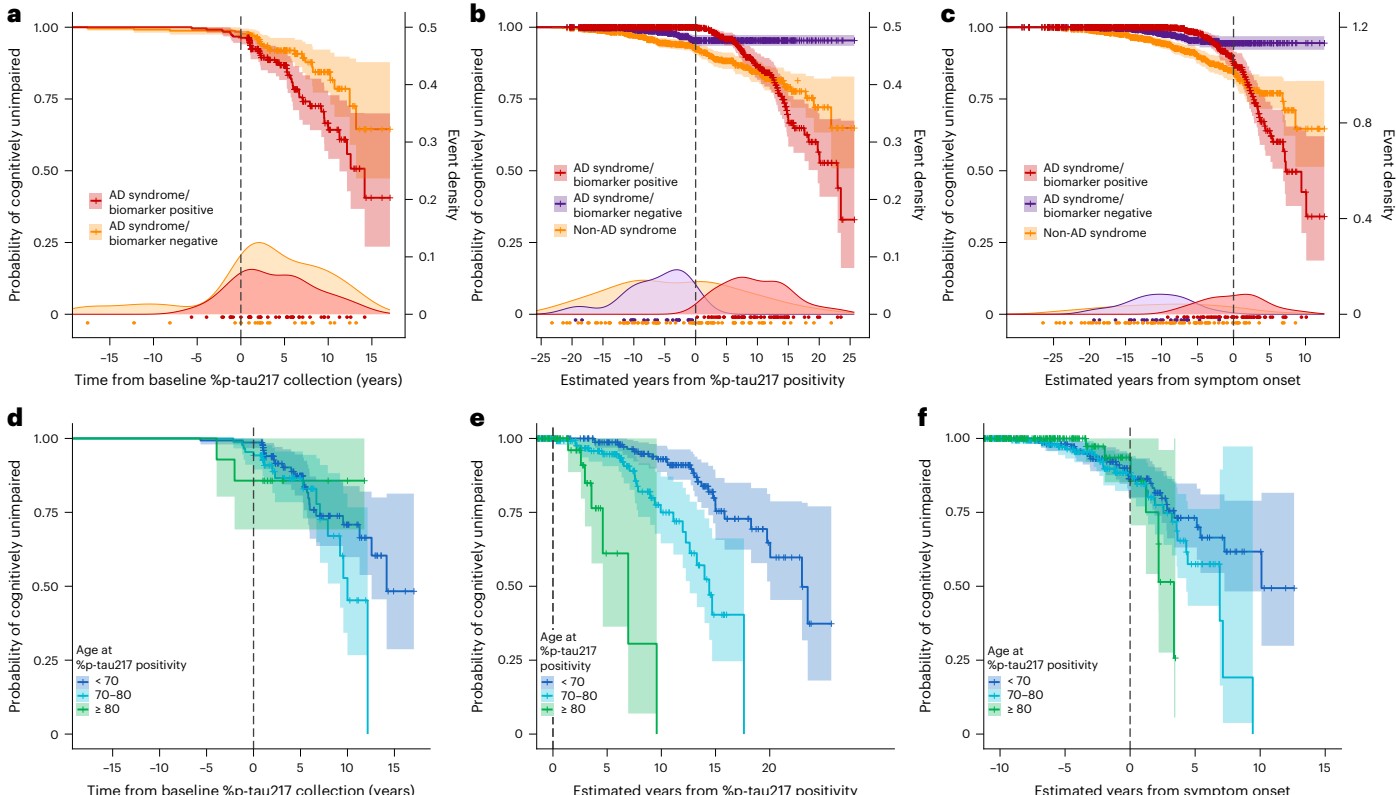

**Fig. 6 | Relationships between cognitive impairment and %p-tau217 positivity.** For initially cognitively unimpaired individuals in the Knight ADRC cohort, the age at plasma %p-tau217 positivity was estimated using the TIRA clock model. Three groups were examined: red (AD syndrome/biomarker positive) had a diagnosis of symptomatic AD at their last assessment with symptoms starting after %p-tau217 positivity; purple (AD syndrome/biomarker negative) had a diagnosis of symptomatic AD at their last assessment with symptoms starting before %p-tau217 positivity; and orange (non-AD syndrome) had a non-AD diagnosis at their last assessment. Kaplan–Meier curves show the probability for each group of remaining cognitively unimpaired individuals as a function of time from first positive %p-tau217 collection (**a**,**d**), estimated years from %p-tau217 positivity (**b**,**e**) or estimated years from symptom onset (**c**,**f**). Density plots and points beneath Kaplan–Meier curves (**a**–**c**) represent the onset of symptoms for individuals in each group. For the red group (AD syndrome/biomarker positive), Kaplan–Meier curves are shown for individuals binned by estimated age of %p-tau217 positivity (blue, <70 years; cyan, 70–80 years; green, ≥80 years) (**d**–**f**).

analysis focused on participants with plasma %p-tau217 values within the interval of consistent change, which increases model reliability but may limit generalizability to individuals with values outside this range. Symptomatic AD was defined as cognitive impairment with an AD syndrome in the context of an estimated positive %p-tau217 value. The threshold for %p-tau217 positivity corresponds to an amyloid PET Centiloid value of 20, below which very few individuals have symptoms due to AD[23,45], making it unlikely that individuals with an estimated negative %p-tau217 value have cognitive impairment due to AD pathology. However, occasional individuals may have discrepant %p-tau217 values. Notably, results were also shown for individuals with an AD syndrome with an estimated negative %p-tau217 value and those with non-AD dementia syndromes. Interpretation of estimates for smaller subgroups or those with mixed clinical presentations may be affected by limited sample sizes and co-pathologies.

Additional limitations include that participants in the study had a variety of clinical diagnoses that were grouped together for analyses, and the models do not reflect the full complexity of clinical symptoms. Participants in the study largely identified as non-Hispanic White, which may limit the generalizability of these models to other groups, especially groups with different rates of non-AD co-pathologies. Furthermore, like other longitudinal aging studies, our analysis did not explicitly model participant dropout or death, which could introduce survival bias if individuals who develop more rapid cognitive decline are more likely to discontinue participation. This potential for survival bias is an important consideration when interpreting our

results, as it may lead to underestimation of decline among the most vulnerable individuals.

In conclusion, our study demonstrates that a single plasma %p-tau217 value can be used to estimate years from onset of AD symptoms with an MdAE of 3–4 years. Models with this level of precision may assist in selecting participants for clinical trials targeting certain phases of preclinical AD. Further refinement of these models could potentially improve predictions, enabling shorter clinical trials and possible relevance for individual decision-making.

## Online content

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

[1]Department of Neurology, Washington University in St. Louis, St. Louis, MO, USA. [2]Northern California Institute for Research and Education, San Francisco, CA, USA. [3]Department of Radiology and Biomedical Imaging, University of California, San Francisco, San Francisco, CA, USA. [4]Wisconsin Alzheimer's Institute, University of Wisconsin-Madison School of Medicine and Public Health, Madison, WI, USA. [5]Department of Medicine, University of Wisconsin-Madison School of Medicine and Public Health, Madison, WI, USA. [6]Rush Alzheimer's Disease Center, Chicago, IL, USA. [7]Department of Neurological Sciences, Rush University Medical Center, Chicago, IL, USA. [8]Knight Alzheimer Disease Research Center, St. Louis, MO, USA. [9]Division of Biostatistics, Washington University School of Medicine, St. Louis, MO, USA. [10]Precision Measures, Johnson & Johnson, San Diego, CA, USA. [11]AbbVie Deutschland GmbH & Co. KG, Ludwigshafen am Rhein, Germany. [12]Takeda Pharmaceutical Company Ltd., Cambridge, MA, USA. [13]Biogen, Cambridge, MA, USA. [14]Alzheimer's Association, Chicago, IL, USA. [15]Department of Pathology and Laboratory Medicine, Perelman School of Medicine, University of Pennsylvania, Philadelphia, PA, USA. [16]Department of Neurology, Indiana University School of Medicine, Indianapolis, IN, USA. [17]Stark Neurosciences Research Institute, Indiana University School of Medicine, Indianapolis, IN, USA. [18]Institute of Neuroscience and Physiology, Department of Psychiatry and Neurochemistry, The Sahlgrenska Academy at University of Gothenburg, Mölndal, Sweden. [19]Banner Alzheimer's Institute, Phoenix, AZ, USA. [20]Banner Sun Health Research Institute, Sun City, AZ, USA. [21]Clinical Neurochemistry Laboratory, Sahlgrenska University Hospital, Mölndal, Sweden. [22]UK Dementia Research Institute Fluid Biomarkers Laboratory, UK DRI at UCL, London, UK. [23]Department of Neurodegenerative Disease, UCL Queen Square Institute of Neurology, London, UK. [24]Hong Kong Center for Neurodegenerative Diseases, Hong Kong, China. [25]Wisconsin Alzheimer's Disease Research Center, University of Wisconsin School of Medicine and Public Health, University of Wisconsin-Madison, Madison, WI, USA. [26]Foundation for the National Institutes of Health, North Bethesda, MD, USA. [27]Department of Psychiatry, Washington University School of Medicine, St. Louis, MO, USA. [28]NeuroGenomics and Informatics Center, Washington University, St. Louis, MO, USA. [29]Hope Center for Neurologic Diseases, St. Louis, MO, USA. [30]Tracy Family SILQ Center for Neurodegenerative Biology, St. Louis, MO, USA. [31]AbbVie, North Chicago, IL, USA. *Lists of authors and their affiliations appear at the end of the paper. ✉e-mail: schindlers@neuro.wustl.edu

**Alzheimer's Disease Neuroimaging Initiative (ADNI)**

Leslie M. Shaw[15]

**On behalf of the Foundation for the National Institutes of Health (FNIH) Biomarkers Consortium Plasma Aβ and Phosphorylated Tau as Predictors of Amyloid and Tau Positivity in Alzheimer's Disease Project Team**

Ziad S. Saad[10], Lei Du-Cuny[11], Janaky Coomaraswamy[12], Yulia Mordashova[11], Carrie E. Rubel[13], Emily A. Meyers[14], Leslie M. Shaw[15], Jeffrey L. Dage[16,17], Henrik Zetterberg[18,21,22,23,24,25], Kyle Ferber[13], Gallen Triana-Baltzer[10], Michael Baratta[12], Erin G. Rosenbaugh[26], J. Martin Sabandal[26], Anthony W. Bannon[31], William Z. Potter & Suzanne E. Schindler[1,8,29]

A full list of members and their affiliations appears in the supplementary information.

## Methods

### Study participants

The STROBE requirements for an observational study were followed. Research participants were included who had been enrolled in previously described studies of memory and aging at the Knight ADRC[16] or the ADNI[13]. Both cohorts consisted of community-dwelling older adults, including participants with and without cognitive impairment, who were followed longitudinally with standardized clinical and biomarker assessments. Sex was self-reported by participants in both cohorts. The Knight ADRC cohort is focused on longitudinal characterization of preclinical AD and the transition to symptomatic AD. ADNI was initiated in 2003 and represents a collaborative effort between public and private sectors, with Michael W. Weiner serving as the principal investigator (https://adni.loni.usc.edu/). The primary goal of the ADNI has been to test whether serial imaging scans, other biological markers and clinical and neuropsychological assessment can be combined to measure the progression of early AD.

All protocols were approved by the Washington University in St. Louis institutional review board (Human Research Protection Office) and by the local institutional review boards at each participating ADNI site. Written informed consent was obtained from every participant or, when appropriate, from a legally authorized representative.

### Plasma biomarkers

Plasma was collected as previously described in the Knight ADRC[16] and ADNI[46,47] cohorts. Plasma %p-tau217 was measured by C2N Diagnostics with a liquid chromatography–mass spectrometry (LC–MS)-based assay[32]. The plasma %p-tau217 measure was calculated as p-tau217 concentration divided by non-phosphorylated tau217 concentration times 100 and is also described as the percent phosphorylation occupancy[27]. The Fujirebio Lumipulse G assay for p-tau217 and Aβ42 was run in singlicate with research-use-only commercially available kits on a Fujirebio Lumipulse G1200 analyzer at the Indiana University National Centralized Repository for Alzheimer's Disease and Related Dementias Biomarker Assay Laboratory (NCRAD-BAL)[23]. The Janssen LucentAD Quanterix and ALZpath Quanterix p-tau217 assays were run in duplicate on a Quanterix Simoa-HD-X analyzer at the Quanterix Accelerator Laboratory. Additional details are included in the study methodology report, available in the ADNI database (https://adni.loni.usc.edu/).

### Clinical and cognitive assessments

Participants underwent clinical assessments that included a detailed interview with a collateral source, a neurological examination of the participant and the CDR[48]. Individuals with CDR = 0 were categorized as 'cognitively unimpaired'. Individuals with CDR > 0 were categorized as 'cognitively impaired'; this group includes individuals with mild cognitive impairment and dementia. Individuals with clinical features consistent with cognitive impairment caused by AD (for example, most commonly, insidious onset, slowly progressive decline and early amnestic impairment but also including logopenic aphasia, posterior cortical dysfunction or dysexecutive presentations) were considered to have an AD syndrome[1,33–35]. Individuals with a primary clinical diagnosis that did not include AD (such as Parkinson disease dementia and vascular dementia) were considered to have a non-AD syndrome. The assessment of clinical syndrome was made by experienced clinicians who were blinded to biomarker results, and determinations were based solely on clinical presentation and established diagnostic criteria and were recorded as the primary clinical diagnosis[1,33–35].

Symptomatic AD was defined to align with the established guidelines for clinical diagnosis of symptomatic AD: cognitive impairment with an AD syndrome in the context of biomarkers, indicating the presence of AD pathology[1,35]. The onset of AD symptoms was defined as the first clinical assessment when initially cognitively unimpaired individuals with positive AD biomarkers (based on estimated %p-tau217) were found to be cognitively impaired with an AD syndrome. Furthermore,

AD symptom onset was applied only to individuals who were cognitively impaired with an AD syndrome at their last assessment—that is, if an individual had transient cognitive impairment but returned to cognitively unimpaired or had a non-AD diagnosis at their last assessment, the earlier impairment was not considered to be the onset of AD symptoms.

For longitudinal visualization and analysis, participants were categorized based on their cognitive status at each assessment and final diagnostic outcome relative to estimated %p-tau217 positivity timing: (1) cognitively unimpaired at the assessment; (2) AD syndrome/ biomarker positive: cognitively impaired at the assessment with a diagnosis of symptomatic AD at their last assessment and symptoms starting after %p-tau217 positivity; (3) AD syndrome/biomarker negative: cognitively impaired at the assessment with a diagnosis of symptomatic AD at their last assessment but symptoms starting before %p-tau217 positivity; and (4) non-AD syndrome: cognitively impaired with a non-AD diagnosis at their last assessment.

### Amyloid and tau PET imaging

Amyloid and tau PET imaging was conducted as previously described[23]. A mesial-temporal meta-region of interest (ROI) that included the entorhinal, parahippocampal and amygdala regions was used to assess early tau pathology ($T_{early}$) with a corresponding positivity threshold of 1.328 standardized uptake value ratio (SUVR) derived from Gaussian mixture modeling using the mean plus 2 s.d. of the first component[23]. A temporo-parietal meta-ROI that included the superior temporal, cuneus, inferior-superior parietal, inferior-middle-superior temporal, isthmus cingulate, lateral occipital, lingual, posterior cingulate, precuneus and superior marginal was used to assess late tau pathology ($T_{late}$) with a corresponding positivity threshold of 1.224 SUVR, using a similar approach for identifying the threshold.

### Cohorts

For analysis of variance in the rate of change in plasma %p-tau217, participants were included who had two or more %p-tau217 values at least 1 year apart. For development of clock models, the cohort was restricted to individuals who had two or more plasma %p-tau217 values between 1.06% and 10.45% at least 1 year apart. For models of age at AD symptom onset, individuals were included who were (1) initially cognitively unimpaired (CDR = 0), (2) subsequently developed cognitive impairment (CDR > 0) with an AD syndrome after estimated plasma %p-tau217 positivity and (3) were cognitively impaired (CDR > 0) with an AD syndrome at their last assessment. For visualization of predicted AD symptom onset as a function of estimated age at plasma %p-tau217 positivity, all individuals or individuals who were cognitively unimpaired at baseline were included.

### Statistical analysis

**Development and validation of plasma %p-tau217 clock models.** Clock models refer to mathematical transformations that convert biomarker levels (for example, plasma %p-tau217) into disease time (estimated years since biomarker positivity), enabling temporal staging of AD pathology progression. This approach first identifies periods of consistent biomarker change and then aligns data relative to time since estimated biomarker positivity rather than chronological age, which complements traditional longitudinal modeling methods. This terminology should be distinguished from general biological aging clocks. Our clock models are developed using the single biomarker %p-tau217, which reflects pathological processes of both amyloid plaques and neurofibrillary tangles[23]. Although these pathologies typically evolve jointly during the progression of AD, our approach does not explicitly model their joint evolution. Instead, by leveraging the strong associations of %p-tau217 with both amyloid and tau PET, our models may capture an integrated measure of disease progression.

The rate of change in plasma %p-tau217 as a function of the estimated %p-tau217 value at the midpoint was modeled with GAMs as previously described[13]. To determine the range of plasma %p-tau217 values over which rates of change were consistent, we quantified prediction uncertainty using squared standard errors from GAMs, which represent the variance of model-estimated rates. As was previously described for models using amyloid and tau PET[13], intervals with variance in rates of change below the 90th percentile were identified. Plasma %p-tau217 values within the interval of consistent change were used for developing clock models.

The TIRA approach estimates individual plasma %p-tau217 rates of change using linear mixed-effects modeling with random slopes and intercepts[10,16]. The rates of change are used in GAMs with cubic splines to characterize nonlinear relationships between the rates of change and plasma %p-tau217 levels at the estimated midpoint of follow-up. The inverse of the modeled rate of change is integrated to derive the time between plasma %p-tau217 values.

The SILA algorithm models longitudinal biomarker trajectories through discrete rate sampling and numerical integration[17]. The method estimates the first-order relationship between biomarker accumulation rate and biomarker levels by sampling rates across evenly distributed values and then applies Euler's method to numerically integrate this relationship into a non-parametric biomarker versus time curve.

Both clocks were centered so that time zero was a plasma %p-tau217 value of 4.06%, which aligns with an amyloid PET Centiloid value of 20 (ref. 23). To obtain an estimated age of plasma %p-tau217 positivity, participants with at least one %p-tau217 value between 1.06% and 10.45% were included. The plasma %p-tau217 clocks were used to calculate an individual's estimated age of %p-tau217 positivity by subtracting the %p-tau217 time from the age at the plasma collection. For example, if an 80-year-old person had a plasma %p-tau217 value of 7.06%, which corresponds to 8.8 years from %p-tau217 positivity (based on the Knight ADRC TIRA clock), their estimated age at %p-tau217 positivity would be 71.2 years (80 years minus 8.8 years). For individuals with more than one plasma %p-tau217 value, their estimated age at %p-tau217 positivity was an average of estimates from all plasma samples.

Scatter plots were generated with data points being color coded by cohort for visualization, to assess the concordance of estimated ages of plasma %p-tau217 positivity between different clock models (TIRA and SILA) and across cohorts (Knight ADRC and ADNI). Associations were evaluated using metrics of adjusted $R^2$, Spearman's $r$ and CCC.

**Modeling the probability of developing symptomatic AD.** The onset of AD symptoms was defined as the first clinical assessment when initially cognitively unimpaired (CDR = 0) individuals with positive AD biomarkers (based on %p-tau217) were found to be cognitively impaired (CDR > 0) with an AD clinical syndrome. For participants who were cognitively normal at baseline, we used interval-censored and right-censored Cox proportional hazards regression models to examine the association between estimated age at plasma %p-tau217 positivity and time to cognitive impairment, accounting for the uncertainty in exact onset timing inherent in longitudinal studies. For participants who remained cognitively unimpaired throughout follow-up, survival times were right-censored at their last assessment age. To account for variable time between assessments, the time of AD symptom onset was interval-censored between the last cognitively unimpaired assessment and the first symptomatic AD assessment. In a sensitivity analysis, participants with cognitive impairment at baseline were included in the models and were left-censored. The models were fitted using the icenReg package in R with semi-parametric baseline hazard estimation. Bootstrap resampling ($n = 5,000$ samples) was performed to obtain robust standard errors and confidence intervals. Model discrimination was assessed using C-indexes calculated specifically for interval-censored data. Survival curves were generated to visualize the probability of remaining cognitively unimpaired across different estimated plasma %p-tau217 positivity age groups. Survival models treated estimated age at %p-tau217 positivity as a fixed covariate from the clock model estimation.

**Estimating the age of AD symptom onset.** For individuals with AD symptom onset after plasma %p-tau217 positivity, linear models estimated AD symptom onset as a function of the estimated age of %p-tau217 positivity. Sensitivity analyses included individuals with AD symptom onset prior to plasma %p-tau217 positivity. Sex, years of education and *APOE* ε4 carrier status were considered as covariates in the models but not included in the final models due to not being significant predictors. Model diagnostics were conducted to ensure the appropriateness of linear modeling. Normality of residuals was assessed using Shapiro−Wilk tests; homoscedasticity was evaluated using Breusch−Pagan tests; and linearity was confirmed through Akaike information criterion (AIC)-based model comparison and $F$-tests comparing linear, quadratic and cubic polynomial specifications. Sensitivity analyses were performed by refitting models after excluding observations with Cook's distance > 4/$n$ to assess model robustness to influential points.

Participant data were visualized as raster plots with estimated age at plasma %p-tau217 positivity on the $y$ axis and estimated years from %p-tau217 positivity on the $x$ axis, with participants ordered by their estimated positivity age and color coded by clinical diagnosis category to illustrate the temporal relationship between estimated %p-tau217 positivity and symptom onset. The estimated age at symptom onset from the linear models was overlaid on the plots to aid visualization of how the timing of symptom onset varies as a function of estimated years since plasma %p-tau217 positivity.

**Examining predicted AD symptom onset across longitudinal cognitive assessments.** To examine cognitive impairment risk across different temporal frameworks, we used Kaplan−Meier curves with different timelines. The primary analysis used time from baseline plasma %p-tau217 collection to onset of cognitive impairment. We also used estimated years from plasma %p-tau217 positivity and estimated years from predicted symptom onset, where predicted symptom onset age was calculated using the linear models described above. Kaplan−Meier curves were combined with density plots showing the distribution of observed events to visualize both survival probabilities and the timing of actual cognitive impairment for each outcome. For each timescale, we binned participants by age at estimated %p-tau217 positivity (<70 years, 70−80 years and ≥80 years) to assess age-dependent risk patterns.

**Alignment of biological AD stages and plasma %p-tau217-derived measures.** Statistical comparisons were performed to evaluate differences in plasma %p-tau217 levels and estimated years from %p-tau217 positivity across the four 2024 Alzheimer's Association biological stages: stage A (normal biomarkers; equivalent to $A-T_{early}-T_{late}-$), stage B (AD pathologic change; equivalent to $A+T_{early}-T_{late}-$), stage C (AD; equivalent to $A+T_{early}+T_{late}-$) and stage D (advanced AD; equivalent to $A+T_{early}+T_{late}+$). Pairwise comparisons between all groups were conducted using non-parametric Conover−Iman tests with Benjamini−Hochberg adjustment for multiple comparisons.

**Development of additional plasma clocks.** The same methodology for developing clocks was implemented using Fujirebio Lumipulse p-tau217/Aβ42 and p-tau217, C2N Diagnostics p-tau217, Janssen LucentAD Quanterix p-tau217 and ALZpath Quanterix p-tau217. Thresholds for each were obtained from previously work[23]. Thresholds were 2.34 pg ml⁻¹ for C2N Diagnostics p-tau217, 0.158 pg ml⁻¹ for Fujirebio Lumipulse p-tau217, 0.0615 pg ml⁻¹ for Janssen LucentAD Quanterix p-tau217 and 0.444 pg ml⁻¹ for ALZpath Quanterix p-tau217.

**Software used.** R version 4.4.1 was used for all analyses except SILA models, which used MATLAB 2024b. Data manipulation and visualization were performed using the 'tidyverse' package for core data handling. Statistical annotations and advanced plot arrangements used the 'ggpubr' package. Publication-ready plot themes were achieved using the 'cowplot' package. Multiple plots were combined and arranged using the 'patchwork' package. Specialized distribution visualization was implemented using the 'ggdist' package. Project management utilities included the 'here' package for project-relative file path management and the 'conflicted' package for function conflict resolution. Parallel computing support was enabled by the 'doParallel' package to enhance computational efficiency.

Statistical modeling employed several specialized packages. Linear mixed-effects modeling was conducted using the 'nlme' package. Generalized additive modeling was implemented using the 'mgcv' package. Interval-censored regression modeling was implemented using the 'icenReg' package. CCC analysis was conducted using the 'DescTools' package. Survival analysis used the 'survival' package with advanced survival plotting provided by the 'survminer' package. Statistical tests and post hoc comparisons were performed with the 'rstatix' package.

### Reporting summary

Further information on research design is available in the Nature Portfolio Reporting Summary linked to this article.

## Data availability

The data that support the findings of this study are not publicly available due to privacy restrictions and participant consent agreements that require controlled access to protect research participant confidentiality. Data from the Knight ADRC can be requested by qualified investigators (https://knightadrc.wustl.edu/professionals-clinicians/request-center-resources/). Data from the ADNI can be requested via the LONI website (https://adni.loni.usc.edu/).

## Code availability

Code developed by the authors for this study is available for download from GitHub: https://github.com/WashUFluidBiomarkers/plasma_ptau217_time. Code for implementing the SILA algorithm is available at https://github.com/Betthauser-Neuro-Lab/SILA-AD-Biomarker.

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

## Acknowledgements

This study represents results of the Foundation for the National Institutes of Health (FNIH; https://fnih.org/) Biomarkers Consortium 'Biomarkers Consortium, Plasma Aβ and Phosphorylated Tau as Predictors of Amyloid and Tau Positivity in Alzheimer's Disease' project, which was made possible through a public–private partnership managed by the FNIH and funded by AbbVie Inc., the Alzheimer's Association, the Diagnostics Accelerator at the Alzheimer's Drug Discovery Foundation, Biogen, Janssen Research & Development LLC and Takeda. We are grateful for the contributions of the following project team members: A. Bannon (AbbVie), M. Baratta (Takeda), J. Coomaraswamy (Takeda), J. Dage (Indiana University), I. Dobler (Takeda), L. Du-Cuny (AbbVie), K. Ferber (Biogen), J. Hsiao (National Institute on Aging (NIA)), H. Kolb (formerly with Johnson & Johnson Innovative Medicine), E. Meyers (Alzheimer's Association), Y. Mordashova (AbbVie), W. Potter, M. Quinton (AbbVie), D. Raunig (Takeda), E. Rosenbaugh (FNIH), C. Rubel (Biogen), Z. Saad (Johnson & Johnson Innovative Medicine), M. Sabandal (FNIH), P. Saletti (Alzheimer's Drug Discovery Foundation), S. Schindler (Washington University in St. Louis), L. Shaw (University of Pennsylvania), G. Triana-Baltzer (Johnson & Johnson Innovative Medicine), C. Weber (Alzheimer's Association) and H. Zetterberg (University of Gothenburg). Funding partners of the project include AbbVie, the Alzheimer's Association, the Diagnostics Accelerator at the Alzheimer's Drug Discovery Foundation, Biogen, Janssen Research & Development and Takeda Pharmaceutical Company. Private sector funding for the study was managed by the FNIH. This study was also supported by NIA grants R01AG070941 (S.E.S.), P30AG066444 (D.M.H.), P01AG003991 (J.C.M.), P01AG026276 (J.C.M.), R01AG067505 (C.X.) and RF1R01AG053550 (C.X.).

Data collection and sharing for this project was funded by the Alzheimer's Disease Neuroimaging Initiative (ADNI) (NIH grant U01 AG024904) and DOD ADNI (Department of Defense award number W81XWH-12-2-0012). The ADNI is funded by the NIA and the National Institute of Biomedical Imaging and Bioengineering and through generous contributions from the following: AbbVie, the Alzheimer's Association; the Alzheimer's Drug Discovery Foundation; Araclon Biotech; BioClinica Inc.; Biogen; Bristol Myers Squibb Company; CereSpir Inc.; Cogstate; Eisai Inc.; Elan Pharmaceuticals Inc.; Eli Lilly and Company; EuroImmun; F. Hoffmann-La Roche Ltd. and its affiliated company, Genentech Inc.; Fujirebio; GE Healthcare; IXICO Ltd.; Janssen Alzheimer Immunotherapy Research & Development LLC; Johnson & Johnson Pharmaceutical Research & Development LLC; Lumosity; Lundbeck; Merck & Co. Inc.; Meso Scale Diagnostics LLC; NeuroRx Research; Neurotrack Technologies; Novartis Pharmaceuticals Corporation; Pfizer Inc.; Piramal Imaging; Servier; Takeda Pharmaceutical Company; and Transition Therapeutics. The Canadian Institutes of Health Research is providing funds to support ADNI clinical sites in Canada. Private sector contributions are facilitated by the FNIH. The grantee organization is the Northern California Institute for Research and Education, and the study is coordinated by the Alzheimer's Therapeutic Research Institute at the University of Southern California. ADNI data are disseminated by the Laboratory for Neuro Imaging at the University of Southern California. A complete listing of ADNI investigators can be found at https://adni.loni.usc.edu/wp-content/uploads/how_to_apply/ADNI_Acknowledgement_List.pdf.0.

## Author contributions

W.Z.P. is a highly qualified expert. K.K.P., M.M.-A., Y.L., L.D.-C., C.X., D.T., B.S., Z.S.S., D.C., J.C., C.E.R., E.A.M., C.M., R.J.B., A.W.B., W.Z.P. and S.E.S. contributed to the conceptualization and study design. K.K.P., M.M.-A., Y.L., L.D.-C., C.X., B.S., L.M.S., J.L.A., N.J.A., H.Z., C.C., E.M., D.M.H., J.C.M. and R.J.B. contributed to data acquisition and curation. K.K.P., M.M.-A. and Y.L. contributed to methodology and software development. K.K.P., M.M.-A., Y.L., L.D.-C., C.X., D.T. and B.S. performed formal analysis. K.K.P., M.M.-A., C.X., D.M.H., R.J.B. and S.E.S. provided resources and supervision. K.K.P. wrote the original draft. All authors edited and approved the final version for publication and agree to be accountable for all aspects of the work.

## Competing interests

K.K.P. has served as a consultant for Eli Lilly and Company. Y.L. is the co-inventor of the technology 'Novel Tau isoforms to predict onset of symptoms and dementia in Alzheimer's disease', which is in the process of licensing by C2N Diagnostics. Z.S.S. and G.T.-B. are employed by Johnson & Johnson Innovative Medicine and may receive salary and stock for their employment. L.D.-C. and Y.M. are employed by AbbVie Deutschland GmbH & Co. KG. J.C. and M.B. receive salary

and company stock as compensation for their employment with Takeda Pharmaceutical Company. C.E.R. and K.F. are employees of and may own stock in Biogen. E.A.M. is employed by the Alzheimer's Association. L.M.S. receives funding from the NIA for ADNI4 and from the NIA for the University of Pennsylvania ADRC P30 for the Biomarker Core. J.L.D. is an inventor on patents or patent applications of Eli Lilly and Company relating to the assays, methods, reagents and/or compositions of matter for P-tau assays and Aβ-targeting therapeutics. J.L.D. has served or is serving as a consultant or on advisory boards for Eisai, AbbVie, Genotix Biotechnologies Inc., Gates Ventures, Gate Neurosciences, Dolby Family Ventures, Karuna Therapeutics, AlzPath Inc., Cognito Therapeutics, Inc., Prevail Therapeutics, Neurogen Biomarking, Spear Bio, the University of Kentucky, Rush University, Tymora and Quanterix. J.L.D. has received research support from ADx Neurosciences, Fujirebio, Roche Diagnostics and Eli Lilly and Company in the past 2 years. J.L.D. has received speaker fees from Eli Lilly and Company and LabCorp. J.L.D. is a founder of and advisor to Monument Biosciences. J.L.D. has stock or stock options in Eli Lilly and Company, Genotix Biotechnologies, AlzPath Inc., Neurogen Biomarking and Monument Biosciences. N.J.A. has received speaking fees from Eli Lilly, Biogen, Quanterix and Alamar Biosciences. H.Z. has served on scientific advisory boards and/or as a consultant for AbbVie, Acumen, Alector, Alzinova, ALZpath, Amylyx, Annexon, Apellis, Artery Therapeutics, AZTherapies, Cognito Therapeutics, CogRx, Denali, Eisai, LabCorp, Merry Life, Nervgen, Novo Nordisk, Optoceutics, Passage Bio, Pinteon Therapeutics, Prothena, Red Abbey Labs, reMYND, Roche, Samumed, Siemens Healthineers, Triplet Therapeutics and Wave; has given lectures sponsored by Alzecure, BioArctic, Biogen, Cellectricon, Fujirebio, Eli Lilly, Novo Nordisk, Roche and WebMD; and is a co-founder of Brain Biomarker Solutions in Gothenburg AB (BBS), which is a part of the GU Ventures Incubator Program (outside submitted work). C.C. has received research support from GlaxoSmithKline and Eisai. C.C. is a member of the scientific advisory board of Circular Genomics and owns stocks. C.C. is also a member of the scientific advisory board of Admit and has served on the scientific advisory boards of GlaxoSmithKline and Novo Nordisk. E.M.M. has participated in speaker engagements for Eisai, Neurology Live and Projects in Knowledge-Kaplan. E.M.M. has had advisory board roles, consulting and data safety monitoring board (DSMB) relationships with Eli Lilly, Alnylam, Alector, Alzamend, Sanofi, AstraZeneca, F. Hoffmann-La Roche, Grifols and Merck. D.M.H. and R.J.B. have equity ownership interest in C2N Diagnostics. D.M.H. and R.J.B. receive income from C2N Diagnostics for serving on the scientific advisory board. D.M.H.

is on the scientific advisory board of Genentech, Denali, Cajal Neurosciences, Acta Pharmaceuticals and Switch and consults for Alnylam, Pfizer and Roche. R.J.B. is a co-inventor on the following US patent applications: 'Methods to detect novel tau species in CSF and use thereof to track tau neuropathology in Alzheimer's disease and other tauopathies' and 'CSF phosphorylated tau and amyloid beta profiles as biomarkers of Tauopathies'. R.J.B. is a co-inventor on a non-provisional patent application: 'Methods of diagnosing and treating based on site-specific tau phosphorylation'. R.J.B. is an unpaid scientific advisory board member of Roche and Biogen and receives research funding from Avid Radiopharmaceuticals, Janssen, Roche/Genentech, Eli Lilly, Eisai, Biogen, AbbVie, Bristol Myers Squibb and Novartis. A.W.B. receives salary and company stock as compensation for his employment with AbbVie. W.Z.P. was previously employed by the National Institute of Mental Health and is a stockholder in Merck & Co. W.Z.P. is a Co-Chair Emeritus for the FNIH Biomarkers Consortium Neuroscience Steering Committee. W.Z.P. serves as a consultant for Karuna, Neurocrine, Neumarker and Vaaji and receives grant support from the NIA along with stock options from Praxis Bioresearch. S.E.S. has served on scientific advisory boards on biomarker testing and education for Eisai and Novo Nordisk and has received speaking fees for presentations on biomarker testing from Eisai, Eli Lilly, Novo Nordisk, Medscape and PeerView. She has provided unpaid scientific advising to Eisai, Johnson & Johnson Innovative Medicine, Eli Lilly, Biogen, Acumen, Cognito Therapeutics and Danaher. M.M.-A., L.D.-C., C.X., D.T., B.S., E.G.R., J.C.M. and J.M.S. have nothing to disclose.

## Additional information

**Extended data** is available for this paper at https://doi.org/10.1038/s41591-026-04206-y.

**Correspondence and requests for materials** should be addressed to Suzanne E. Schindler.

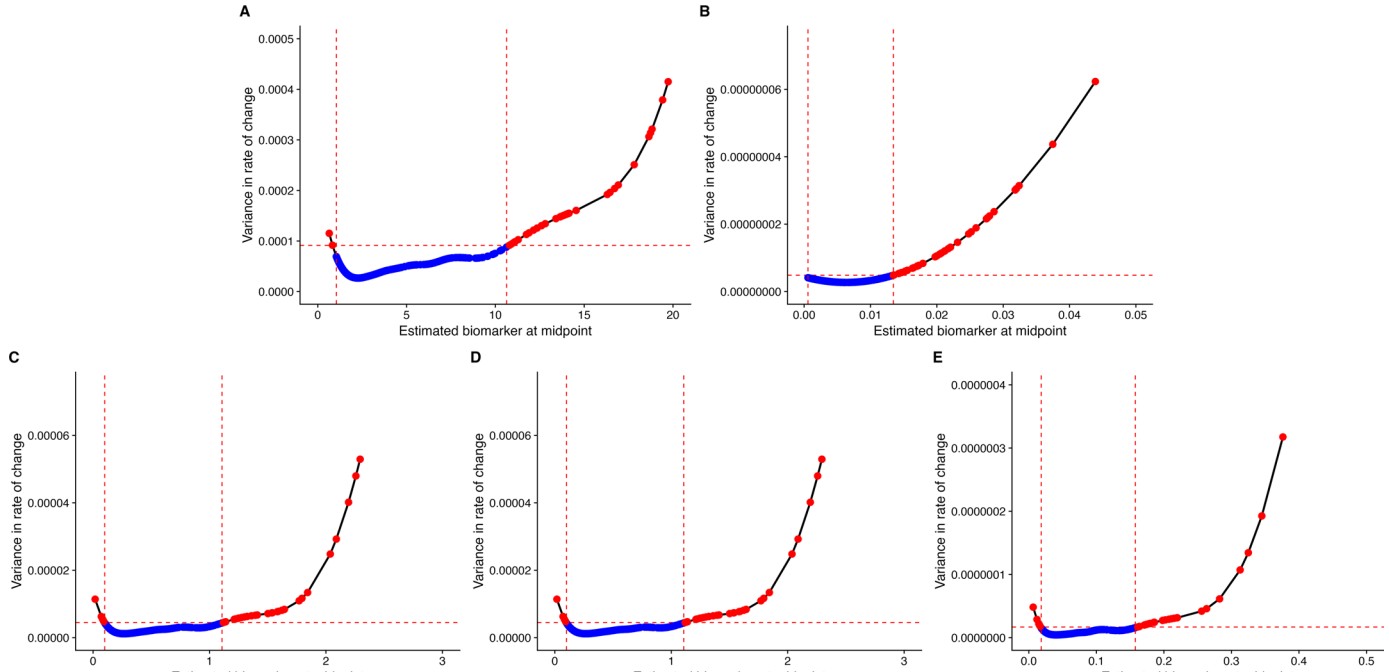

**Extended Data Fig. 1 | Variance analysis for plasma p-tau217 biomarkers identifies intervals of consistent change.** Variance analysis was performed to identify biomarker ranges with relatively consistent rates of change (below the 90th percentile) for five plasma p-tau217 assays: C2N PrecivityAD2 %p-tau217 (**A**), Fujirebio Lumipulse p-tau217/Aβ42 (**B**), Fujirebio Lumipulse p-tau217 (**C**), ALZpath Quanterix p-tau217 (**D**), and Janssen LucentAD Quanterix p-tau217 (**E**). The rate of change in each biomarker was modeled using Generalized Additive Models (GAMs), and variance in the rate of change was calculated across biomarker values. Red dashed vertical lines mark the boundaries of intervals where variance in rates of change was below the 90th percentile, defining the biomarker ranges suitable for clock model development. Blue points indicate intervals with variance below the 90th percentile; red points indicate intervals with variance above the 90th percentile.

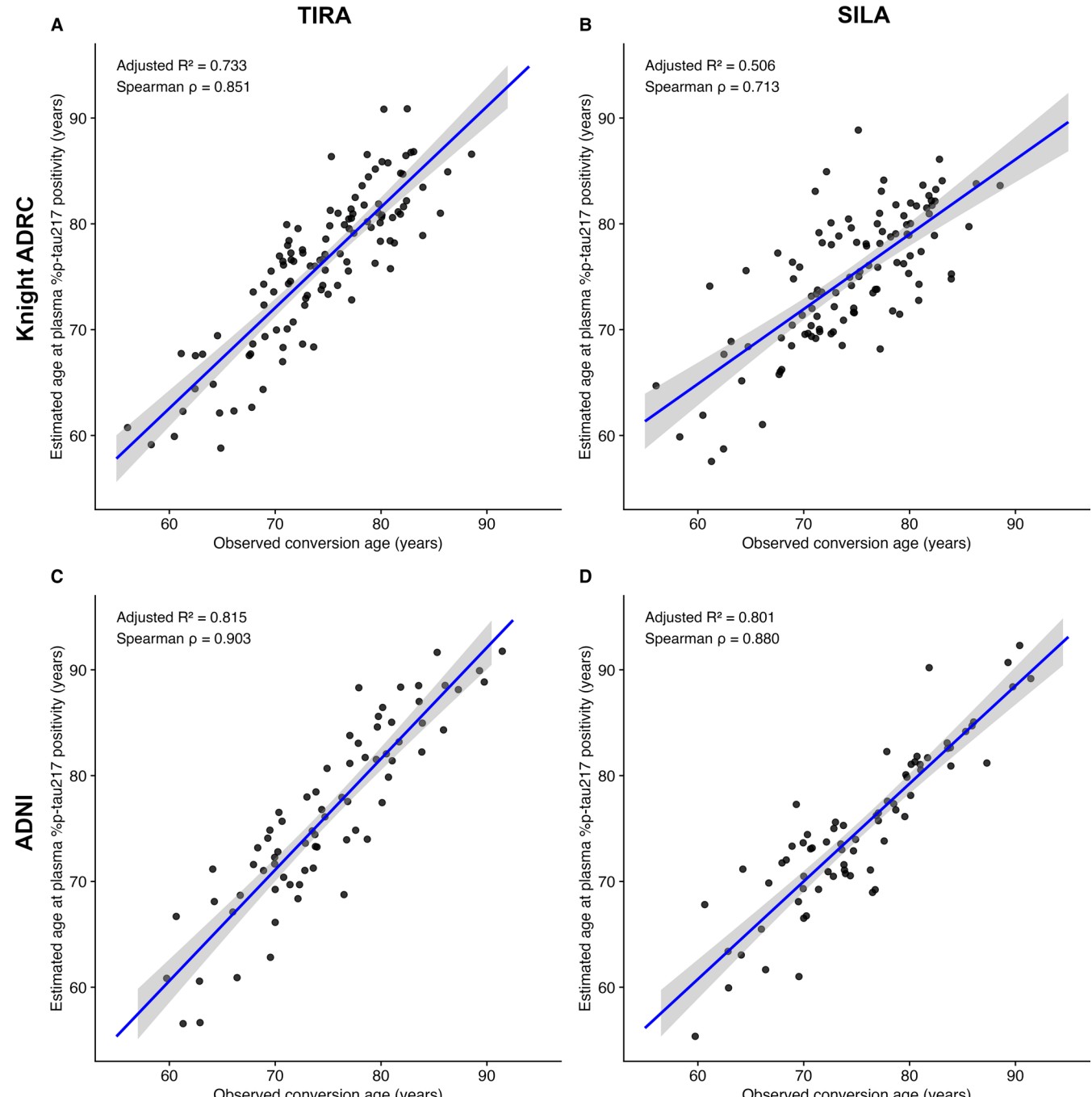

**Extended Data Fig. 2 | Relationship between estimated age at plasma %p-tau217 positivity and age of observed conversion to %p-tau217 positive.** The estimated age at plasma %p-tau217 positivity based on clock models is shown as a function of the observed conversion age, which is the average of the age at last negative and first positive %p-tau217. Clock models in the Knight ADRC (A, B) and ADNI cohorts (C, D) were created with the TIRA (A, C) and SILA (B, D) approaches. Each point represents an individual participant. Blue lines represent linear regression fits, and gray shaded bands represent 95% confidence intervals for the regression line. Adjusted $R^2$ and Spearman correlation coefficients ($\rho$) are shown for each model.

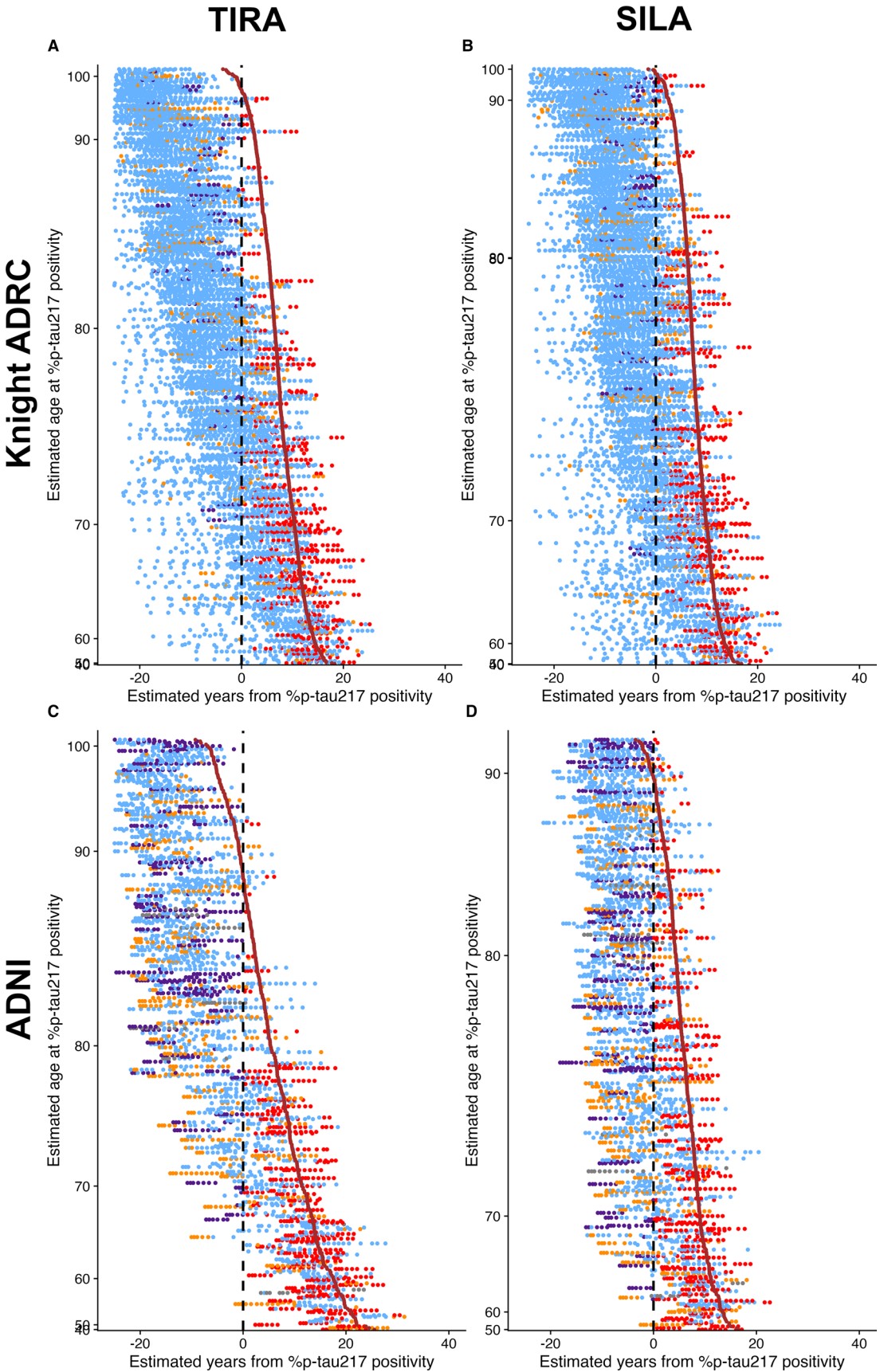

**Extended Data Fig. 3 | See next page for caption.**

**Extended Data Fig. 3 | Clinical diagnosis as a function of estimated age at plasma %p-tau217 positivity and years from %p-tau217 positivity.** For individuals in the Knight ADRC (**A, B**) and ADNI (**C, D**) cohorts, age at plasma %p-tau217 positivity was estimated using either the TIRA (**A, C**) or SILA (**B, D**) models. Each row represents the longitudinal clinical diagnoses for one individual by estimated years from %p-tau217 positivity (x-axis). Individuals are sorted vertically by estimated age at %p-tau217 positivity (y-axis). The point color denotes the clinical diagnosis: blue was cognitively unimpaired at the assessment; red (AD syndrome/biomarker positive) was cognitively impaired at the assessment and had a diagnosis of symptomatic AD at their last assessment with symptoms starting after %p-tau217 positivity; purple (AD syndrome/ biomarker negative) was cognitively impaired at the assessment and had a diagnosis of symptomatic AD at their last assessment with symptoms starting before %p-tau217 positivity; orange (non-AD syndrome) was cognitively impaired and had a non-AD diagnosis at their last assessment. Vertical dashed lines at 0 represent the estimated time of %p-tau217 positivity. Brown lines indicate the relationship between %p-tau217 positivity age and estimated symptom onset based on the Knight ADRC models in Fig. 4.

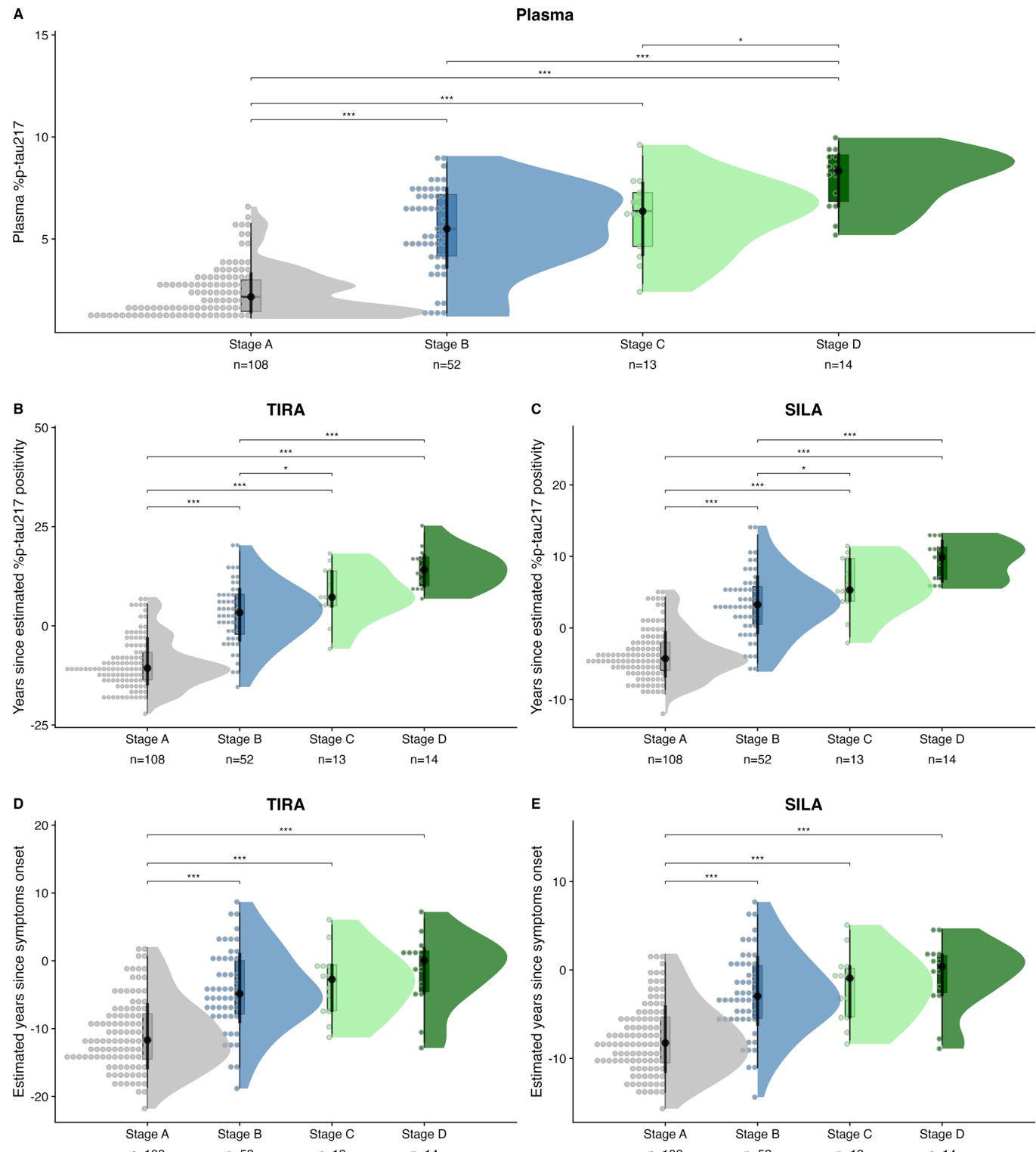

**Extended Data Fig. 4 | Plasma %p-tau217 level, estimated years since %p-tau217 positivity, and years since estimated symptom onset across Alzheimer's Association biological stages.** Using data from ADNI, raincloud plots show plasma %p-tau217 levels (**A**), estimated years since %p-tau217 positivity by TIRA (**B**) and SILA (**C**), and years since estimated symptom onset by TIRA (**D**) and SILA (**E**) stratified by the 2024 Alzheimer's Association biological staging framework: Stage A (normal biomarkers), Stage B (Alzheimer's disease pathologic change), Stage C (Alzheimer's disease), and Stage D (advanced Alzheimer's disease). The n values provided for each group indicate the number of individual human participants included in that biological stage; each participant contributes one data point to the analysis. Raincloud plots display individual data points, probability distributions (violin plots), and box plots where the box represents the interquartile range (25th to 75th percentile), the center line represents the median, and whiskers extend to the minimum and maximum values within 1.5× the interquartile range; points beyond whiskers represent outliers. Group means with 95% confidence intervals are shown. Statistical comparisons between all biological stages were conducted using two-sided non-parametric Conover-Iman tests with Benjamini-Hochberg adjustment for multiple comparisons. Significant group differences are indicated by asterisks (*p < 0.05, **p < 0.01, ***p < 0.001).

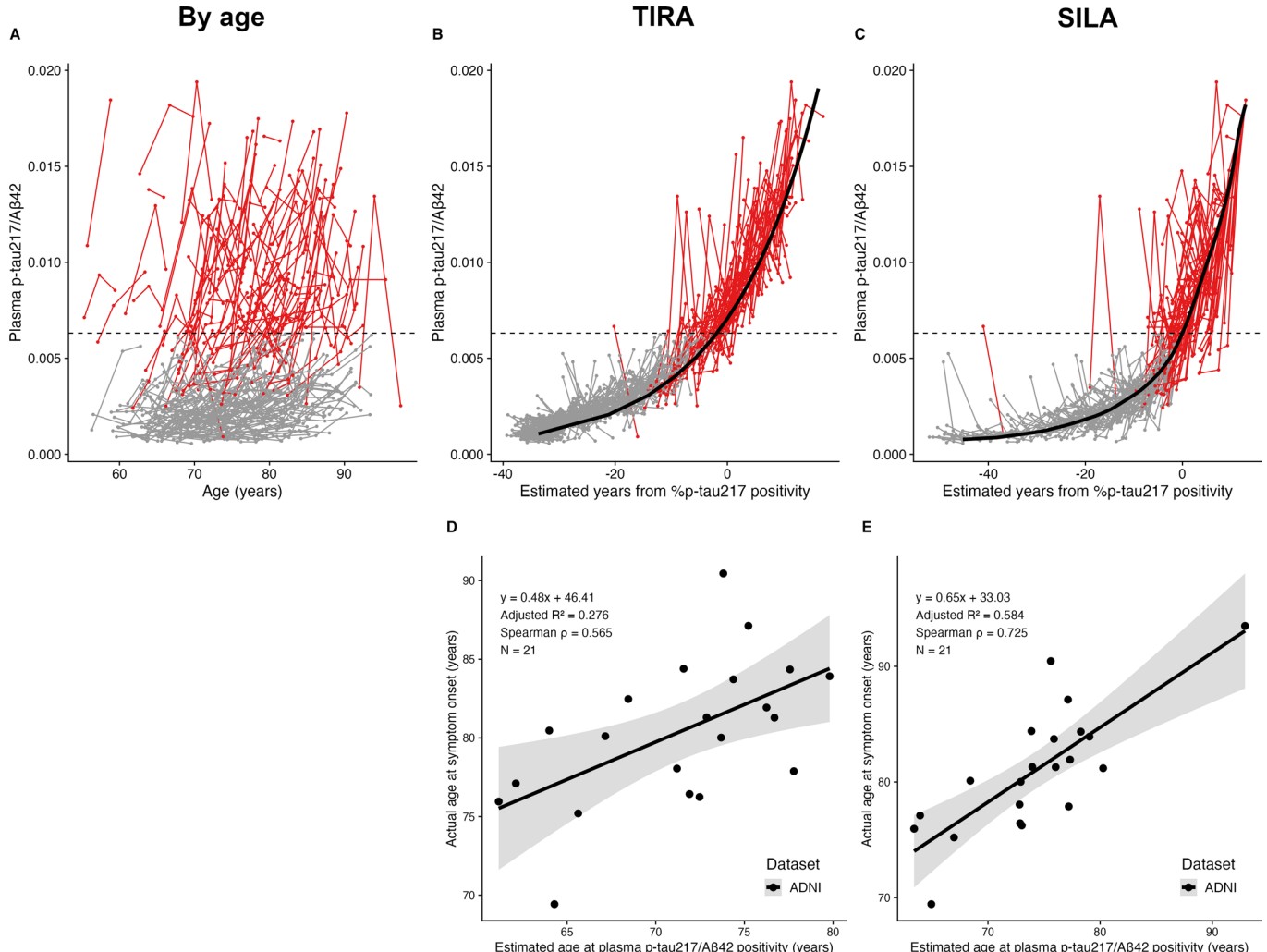

**Extended Data Fig. 5 | Fujirebio Diagnostics' Lumipulse plasma p-tau217/Aβ42 clock and symptom onset models based on estimated years from p-tau217/ Aβ42 positivity.** Longitudinal plasma p-tau217/Aβ42 data from ADNI is shown as a function of age (**A**) or estimated years from p-tau217/Aβ42 positivity by TIRA (**B**) or SILA clock models (**C**). Thick black lines represent the clock models, red lines represent individuals with at least one plasma p-tau217/Aβ42 > 0.006312, and grey lines represent individuals with no plasma p-tau217/Aβ42 > 0.006312. The horizontal black dashed lines represent the plasma p-tau217/Aβ42 threshold of 0.006312. Models for age at symptom onset included individuals who were initially cognitively unimpaired but had a typical AD syndrome at their last assessment and developed symptoms after estimated plasma p-tau217/Aβ42 positivity. Age at p-tau217/Aβ42 positivity was estimated using TIRA (**D**) or SILA (**E**) models. Each point represents an individual participant. In D and E, black lines represent linear regression fits showing the predicted mean age at symptom onset, and gray shaded bands indicate 95% confidence intervals around the regression lines. Linear regression equations, adjusted $R^2$ values, Spearman correlation coefficients (ρ), and sample sizes (N) are shown.

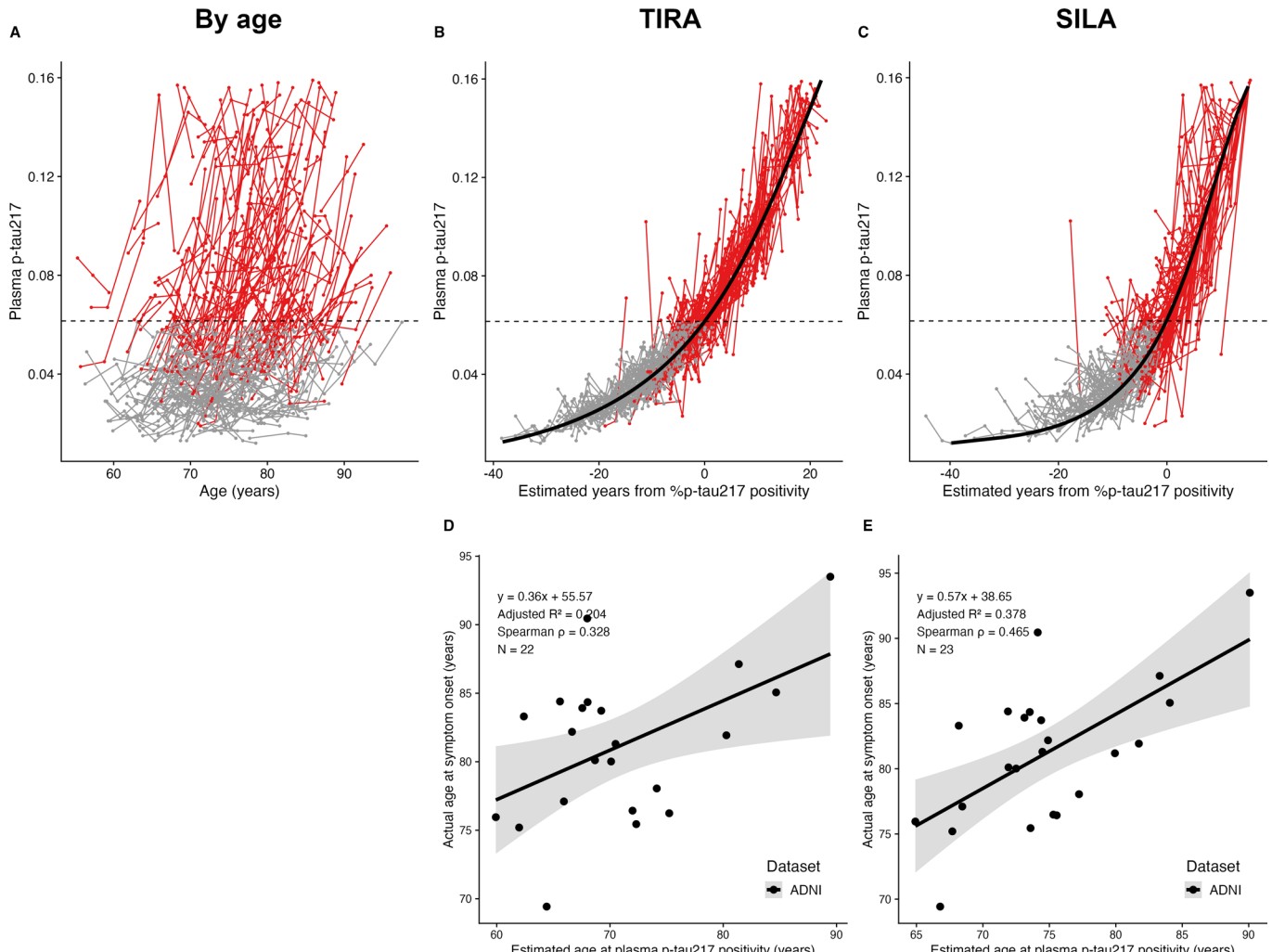

**Extended Data Fig. 6 | Janssen's LucentAD Quanterix plasma p-tau217 and symptom onset models based on estimated years from p-tau217 positivity.** Longitudinal plasma p-tau217 data from ADNI is shown as a function of age (**A**) or estimated years from p-tau217 positivity by TIRA (**B**) or SILA clock models (**C**). Thick black lines represent the clock models; red lines represent individuals with at least one plasma p-tau217 > 0.0615 pg ml⁻¹ and grey lines represent individuals with no plasma p-tau217 > 0.0615 pg ml⁻¹. The horizontal black dashed lines represent the plasma p-tau217 threshold of 0.0615 pg ml⁻¹. Models for age at symptom onset included individuals who were initially cognitively unimpaired but had a typical AD syndrome at their last assessment and developed symptoms after estimated plasma p-tau217 positivity. Age at p-tau217 positivity was estimated using TIRA (**D**) or SILA (**E**) models. Each point represents an individual participant. In D and E, black lines represent linear regression fits showing the predicted mean age at symptom onset, and gray shaded bands indicate 95% confidence intervals around the regression lines. Linear regression equations, adjusted $R^2$ values, Spearman correlation coefficients (ρ), and sample sizes (N) are shown.

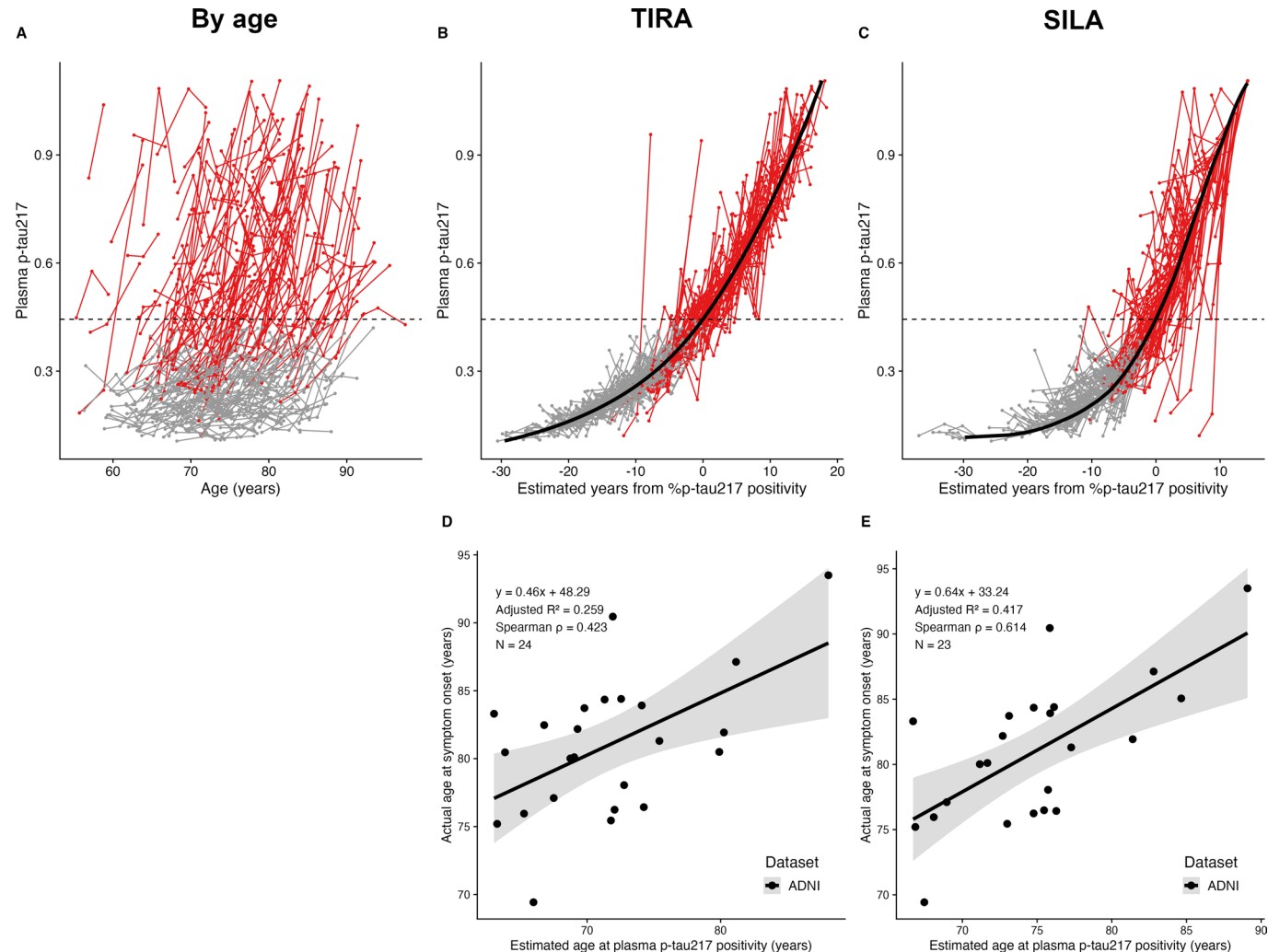

**Extended Data Fig. 7 | ALZpath Quanterix plasma p-tau217 and symptom onset models based on estimated years from p-tau217 positivity.** Longitudinal plasma p-tau217 data from ADNI is shown as a function of age (**A**) or estimated years from p-tau217 positivity by TIRA (**B**) or SILA clock models (**C**). Thick black lines represent the clock models; red lines represent individuals with at least one plasma p-tau217 > 0.444 pg ml$^{-1}$ and grey lines represent individuals with no plasma p-tau217 > 0.444 pg ml$^{-1}$. The horizontal black dashed lines represent the plasma p-tau217 threshold of 0.444 pg ml$^{-1}$. Models for age at symptom onset included individuals who were initially cognitively unimpaired but had a typical AD syndrome at their last assessment and developed symptoms after estimated plasma p-tau217 positivity. Age at p-tau217 positivity was estimated using TIRA (**D**) or SILA (**E**) models. Each point represents an individual participant. In D and E, black lines represent linear regression fits showing the predicted mean age at symptom onset, and gray shaded bands indicate 95% confidence intervals around the regression lines. Linear regression equations, adjusted R$^2$ values, Spearman correlation coefficients (ρ), and sample sizes (N) are shown.

**Extended Data Table 1 | Characteristics of clock model development cohorts**

| Characteristic | | Full Cohort | | Knight ADRC | | ADNI | |
|---|---|---|---|---|---|---|---|
| | n | Values | n | Values | n | Values | p= |
| Age (Years) | 603 | 71.2 (67-76.7) | 258 | 70.0 (65.8-74.2) | 345 | 72.7 (67.8-78.1) | <0.0001 |
| Sex (% Female) | 603 | 300, 49.8% | 258 | 133, 51.6% | 345 | 167, 48.4% | 0.40 |
| *APOE* ε4 carriers (% carrier) | 603 | 217, 36% | 258 | 106, 41.1% | 345 | 111, 32.2% | 0.02 |
| Years of education | 603 | 16 (14-18) | 258 | 16 (15-18) | 345 | 16 (14-18) | 0.53 |
| Cognitive impairment (% CDR>0) | 603 | 179, 29.7% | 258 | 18, 7.0% | 345 | 161, 46.7% | <0.0001 |
| CDR Sum of Boxes | 603 | 0 (0-0.5) | 258 | 0 (0-0) | 345 | 0 (0-1) | <0.0001 |
| Estimated age of %-ptau217 positivity (Years) | 603 | 79.8 (72.1-86.5) | 258 | 78.8 (72.2-83.5) | 345 | 81.0 (71.5-88.5) | 0.0070 |
| Follow-up time (Years) | 603 | 5.5 (4.0-7.5) | 258 | 6.5 (3.9-9.8) | 345 | 4.5 (4.0-6.3) | <0.0001 |
| Number of %p-tau217 values included | 603 | 3 (2-3) | 258 | 2 (2-3) | 345 | 3 (3-3) | <0.0001 |
| First %p-tau217 included | 603 | 2.33 (1.54-4.12) | 258 | 2.22 (1.58-3.44) | 345 | 2.49 (1.53-4.74) | 0.068 |

Individuals were included who had two or more plasma %p-tau217 measurements between 1.06% to 10.45% that were at least one year apart. Values are presented as the median (interquartile range) for continuous variables and n (%) for categorical variables. The significance of differences between cohorts was evaluated by Chi-square tests for categorical values and Wilcoxon ranked sum tests for continuous values.

**Extended Data Table 2 | Clock models estimating time between plasma %p-tau217 levels**

| | | TIRA | | | | SILA | | | |
|---|---|---|---|---|---|---|---|---|---|
| | | Knight ADRC | | ADNI | | Knight ADRC | | ADNI | |
| Beginning %p-tau217 | Ending %p-tau217 | Interval | Cumulative (from 4.06%) | Interval | Cumulative (from 4.06%) | Interval | Cumulative (from 4.06%) | Interval | Cumulative (from 4.06%) |
| 1.06 | 2.06 | 10.3 | -10.7 | 10.9 | -12 | 5.0 | -7.0 | 6.5 | -6.3 |
| 2.06 | 3.06 | 6.2 | -4.5 | 6.9 | -5.1 | 4.0 | -3.0 | 3.5 | -2.8 |
| 3.06 | 4.06 | 4.5 | 0 | 5.1 | 0 | 3.0 | 0 | 2.8 | 0 |
| 4.06 | 5.06 | 3.5 | 3.5 | 4.3 | 4.3 | 2.5 | 2.5 | 2.8 | 2.8 |
| 5.06 | 6.06 | 2.9 | 6.4 | 3.9 | 8.2 | 2.5 | 5.0 | 2.5 | 5.3 |
| 6.06 | 7.06 | 2.4 | 8.8 | 3.7 | 11.9 | 2.2 | 7.2 | 2.0 | 7.3 |
| 7.06 | 8.06 | 2.1 | 10.9 | 3.7 | 15.6 | 2.2 | 9.4 | 2.2 | 9.5 |
| 8.06 | 9.06 | 1.9 | 12.8 | 3.8 | 19.4 | 2.0 | 11.4 | 2.5 | 12.0 |
| 9.06 | 10.06 | 1.7 | 14.5 | 3.9 | 23.3 | 1.8 | 13.2 | 2.5 | 14.5 |
| 10.06 | 10.40 | 0.6 | 15.1 | 1.2 | 24.5 | 0.5 | 13.7 | 0.5 | 15.0 |

The time in years was estimated for plasma %p-tau217 values from 1.06% to 10.40%. Data are shown for two methods (TIRA and SILA) across two cohorts (Knight ADRC and ADNI). Both the interval time between adjacent levels and the cumulative time from the 4.06% reference point are shown and demonstrate differences across methods and cohorts. Negative time intervals correspond to times before %p-tau217 positivity.

**Extended Data Table 3 | Validation of estimated age at symptom onset**

| Check Type | Model cohort | Test cohort | Method | No. of converters | Spearman correlation | Adjusted R² | CCC | RMSE | MAE | MdAE |
|---|---|---|---|---|---|---|---|---|---|---|
| Internal validation (within cohort) | Knight ADRC | Knight ADRC | TIRA | 59 | 0.752 | 0.599 | 0.839 | 4.3 | 3.5 | 3.1 |
| | Knight ADRC | Knight ADRC | SILA | 61 | 0.765 | 0.612 | 0.837 | 4.1 | 3.4 | 3.1 |
| | ADNI | ADNI | TIRA | 20 | 0.522 | 0.337 | 0.771 | 4.4 | 3.8 | 3.5 |
| | ADNI | ADNI | SILA | 22 | 0.673 | 0.470 | 0.814 | 3.7 | 3.2 | 3.0 |
| External validation (independent cohort) | Knight ADRC | ADNI | TIRA | 20 | 0.595 | 0.467 | 0.805 | 4.0 | 3.3 | 3.0 |
| | Knight ADRC | ADNI | SILA | 22 | 0.611 | 0.463 | 0.801 | 4.0 | 3.5 | 3.2 |
| | ADNI | Knight ADRC | TIRA | 59 | 0.669 | 0.509 | 0.808 | 4.9 | 4.0 | 3.6 |
| | ADNI | Knight ADRC | SILA | 62 | 0.736 | 0.577 | 0.820 | 4.5 | 3.8 | 3.7 |

Metrics are shown for linear models estimating the age at symptom onset based on the estimated age at %p-tau217 positivity using either TIRA or SILA clocks. Metrics are shown for internal validation (same model and test cohort) and external validation (different model and test cohort): Spearman correlation, adjusted $R^2$, non-parametric concordance correlation coefficient (CCC), root mean squared error (RMSE), mean absolute error (MAE), and median absolute error (MdAE).

# Reporting Summary

## Statistics

For all statistical analyses, confirm that the following items are present in the figure legend, table legend, main text, or Methods section.

| n/a | Confirmed | |
|---|---|---|
| ☐ | ☒ | The exact sample size (*n*) for each experimental group/condition, given as a discrete number and unit of measurement |
| ☐ | ☒ | A statement on whether measurements were taken from distinct samples or whether the same sample was measured repeatedly |
| ☐ | ☒ | The statistical test(s) used AND whether they are one- or two-sided<br>*Only common tests should be described solely by name; describe more complex techniques in the Methods section.* |
| ☐ | ☒ | A description of all covariates tested |
| ☐ | ☒ | A description of any assumptions or corrections, such as tests of normality and adjustment for multiple comparisons |
| ☐ | ☒ | A full description of the statistical parameters including central tendency (e.g. means) or other basic estimates (e.g. regression coefficient) AND variation (e.g. standard deviation) or associated estimates of uncertainty (e.g. confidence intervals) |
| ☐ | ☒ | For null hypothesis testing, the test statistic (e.g. *F*, *t*, *r*) with confidence intervals, effect sizes, degrees of freedom and *P* value noted<br>*Give P values as exact values whenever suitable.* |
| ☒ | ☐ | For Bayesian analysis, information on the choice of priors and Markov chain Monte Carlo settings |
| ☐ | ☒ | For hierarchical and complex designs, identification of the appropriate level for tests and full reporting of outcomes |
| ☐ | ☒ | Estimates of effect sizes (e.g. Cohen's *d*, Pearson's *r*), indicating how they were calculated |

*Our web collection on statistics for biologists contains articles on many of the points above.*

## Software and code

Policy information about availability of computer code

| | |
|---|---|
| Data collection | Plasma %p-tau217 was measured by C2N Diagnostics using liquid chromatography mass spectrometry (LC-MS) based assay. Additional assays used Fujirebio Lumipulse G1200 analyzer and Quanterix Simoa-HD-X analyzer. Clinical assessments used standardized protocols from Knight ADRC and ADNI. |
| Data analysis | R version 4.4.1 was used for all analyses except SILA models, which used Matlab 2024b. Key R packages included: tidyverse (data manipulation, version 2.0.0), mgcv (GAMs, version 1.9.3), icenReg (interval-censored regression, version 2.0.16), nlme (mixed-effects modeling, version 3.1.168), survival (survival analysis, version 3.8.3), DescTools (concordance correlation, version 0.99.60), doParallel (parallel computing, version 1.0.17). Code available at: https://github.com/WashU-FluidBiomarkers/plasma-ptau217-time |

For manuscripts utilizing custom algorithms or software that are central to the research but not yet described in published literature, software must be made available to editors and reviewers. We strongly encourage code deposition in a community repository (e.g. GitHub). See the Nature Portfolio guidelines for submitting code & software for further information.

## Data

Policy information about availability of data

All manuscripts must include a data availability statement. This statement should provide the following information, where applicable:
- Accession codes, unique identifiers, or web links for publicly available datasets
- A description of any restrictions on data availability
- For clinical datasets or third party data, please ensure that the statement adheres to our policy

Data from the Knight ADRC can be requested by qualified investigators (knightadrc.wustl.edu/Research/ResourceRequest.htm). Data from ADNI can be requested via the LONI website (adni.loni.usc.edu). Code developed by the authors is available at https://github.com/WashU-FluidBiomarkers/plasma-ptau217-time

## Research involving human participants, their data, or biological material

Policy information about studies with human participants or human data. See also policy information about sex, gender (identity/presentation), and sexual orientation and race, ethnicity and racism.

| | |
|---|---|
| Reporting on sex and gender | Sex was reported as a biological attribute in both cohorts. In the full longitudinal cohort (n=912), 52% were female. Knight ADRC had 54.2% female participants (274/506) and ADNI had 49.3% female participants (200/406). Sex was considered as a covariate in symptom onset models but was not a significant predictor and thus not included in final models. Complete sex distributions are provided in Supplementary Tables 1, 2, 6, and 7 for all analysis cohorts. |
| Reporting on race, ethnicity, or other socially relevant groupings | Participants largely identified as non-Hispanic White, which may limit the generalizability of these models to other groups, especially groups with different rates of non-AD co-pathologies. This limitation in demographic diversity is acknowledged in the manuscript's discussion section. |
| Population characteristics | Detailed demographic characteristics are provided in Supplementary Tables 1, 2, 6, and 7. Key baseline characteristics include: median age 69.8 years (Knight ADRC: 67.7 years, ADNI: 72.7 years), 35.1% APOE ε4 carriers, median education 16 years. Cognitive status varied by cohort: Knight ADRC had 8.5% cognitively impaired at baseline vs 48.3% in ADNI. Follow-up time ranged from 4-14 years with multiple longitudinal assessments. |
| Recruitment | Participants were community-dwelling older adults enrolled in established longitudinal aging studies. Knight ADRC participants were recruited through the Knight Alzheimer Disease Research Center at Washington University, focused on characterizing preclinical AD transitions. ADNI participants were recruited through the multi-center Alzheimer's Disease Neuroimaging Initiative, representing a collaborative public-private partnership. Both cohorts used standardized protocols for clinical and biomarker assessments. No specific selection biases are reported, though the predominantly non-Hispanic White demographic may limit generalizability. |
| Ethics oversight | All participants provided written informed consent. The study followed STROBE requirements for observational studies. Research was conducted through established IRB-approved protocols at the Knight Alzheimer Disease Research Center (Washington University) and the Alzheimer's Disease Neuroimaging Initiative consortium. |

Note that full information on the approval of the study protocol must also be provided in the manuscript.

# Field-specific reporting

Please select the one below that is the best fit for your research. If you are not sure, read the appropriate sections before making your selection.

☒ Life sciences          ☐ Behavioural & social sciences          ☐ Ecological, evolutionary & environmental sciences

For a reference copy of the document with all sections, see nature.com/documents/nr-reporting-summary-flat.pdf

# Life sciences study design

All studies must disclose on these points even when the disclosure is negative.

| | |
|---|---|
| Sample size | Sample sizes were determined by the availability of participants with longitudinal plasma %p-tau217 measurements in two established cohorts. The Knight ADRC clock cohort included 258 individuals and the ADNI clock cohort included 345 individuals. For symptom onset models, sample sizes were: Knight ADRC (59-61 individuals) and ADNI (20-22 individuals). No formal power calculations were performed as this was an exploratory analysis using available longitudinal data from established aging cohorts. Sample sizes are reported for each analysis in the main text and Supplementary Tables 1 and 2. |
| Data exclusions | Participants were excluded if they had fewer than two plasma %p-tau217 measurements at least one year apart. For clock model development, analysis was restricted to individuals with plasma %p-tau217 values between 1.06-10.45, representing the range with consistent rates of change identified through variance analysis. Values outside this range showed high variance or sparse data that would make time estimates unstable. For symptom onset models, individuals were excluded if they were cognitively impaired at baseline, had transient cognitive impairment that resolved, or had non-AD diagnoses at final assessment. |
| Replication | The study employed multiple forms of replication: (1) Cross-cohort validation using two independent datasets (Knight ADRC and ADNI), (2) Cross-validation where models trained in one cohort were tested in the other, achieving moderate associations (adjusted R² 0.463-0.577), (3) |

| | |
|---|---|
| | Two different mathematical approaches (TIRA and SILA) that yielded similar results, providing methodological replication, and (4) Validation across multiple plasma p-tau217 assays (5 different commercial assays tested). |
| Randomization | Not applicable. This was an observational longitudinal study of aging cohorts. Participants were not randomized to groups. Clinical assessments and plasma collections followed standardized protocols in both cohorts at predetermined intervals. |
| Blinding | Clinicians performing clinical assessments and determining cognitive diagnoses were blinded to biomarker results. Clinical syndrome determinations (AD vs non-AD) were made based solely on clinical presentation and established diagnostic criteria without knowledge of plasma %p-tau217 values or other biomarker results. |

# Reporting for specific materials, systems and methods

We require information from authors about some types of materials, experimental systems and methods used in many studies. Here, indicate whether each material, system or method listed is relevant to your study. If you are not sure if a list item applies to your research, read the appropriate section before selecting a response.

## Materials & experimental systems

| n/a | Involved in the study |
|---|---|
| ☒ ☐ | Antibodies |
| ☒ ☐ | Eukaryotic cell lines |
| ☒ ☐ | Palaeontology and archaeology |
| ☒ ☐ | Animals and other organisms |
| ☐ ☒ | Clinical data |
| ☒ ☐ | Dual use research of concern |
| ☒ ☐ | Plants |

## Methods

| n/a | Involved in the study |
|---|---|
| ☒ ☐ | ChIP-seq |
| ☒ ☐ | Flow cytometry |
| ☒ ☐ | MRI-based neuroimaging |

## Clinical data

Policy information about clinical studies

All manuscripts should comply with the ICMJE guidelines for publication of clinical research and a completed CONSORT checklist must be included with all submissions.

| | |
|---|---|
| Clinical trial registration | Not applicable. This was an observational longitudinal study using existing cohorts (Knight ADRC and ADNI), not a clinical trial requiring registration. Both studies follow established longitudinal protocols approved by their respective institutional review boards. |
| Study protocol | Full study protocols are available through the respective cohort websites. Knight ADRC protocols can be accessed through https://knightadrc.wustl.edu/Research/ResourceRequest.htm. ADNI protocols are available at https://adni.loni.usc.edu. Both cohorts use standardized protocols for clinical assessments, biomarker collection, and longitudinal follow-up that have been previously published and are publicly documented. |
| Data collection | Data were collected at two sites: Knight Alzheimer Disease Research Center at Washington University in St. Louis (single-site cohort) and multiple centers participating in the Alzheimer's Disease Neuroimaging Initiative (ADNI, multi-center consortium). Plasma samples were collected during routine study visits following standardized protocols. Knight ADRC participants had plasma collected over a median of 7.1 years (2012-2024 timeframe), while ADNI participants had plasma collected over a median of 5.0 years (2005-2021 timeframe). Clinical assessments were conducted at regular intervals using standardized protocols including Clinical Dementia Rating (CDR) evaluations and neurological examinations. |
| Outcomes | Primary outcome was age at onset of symptomatic Alzheimer disease, defined as the first clinical assessment when initially cognitively unimpaired (CDR=0) individuals with positive AD biomarkers developed cognitive impairment (CDR>0) with an AD syndrome. Secondary outcomes included probability of developing symptomatic AD over time and cognitive impairment as measured by the Preclinical Alzheimer Cognitive Composite (PACC). Clinical syndrome determinations were made by clinicians blinded to biomarker results using established diagnostic criteria. All assessments followed standardized protocols with interval-censored timing to account for variable assessment intervals. |

## Plants

| | |
|---|---|
| Seed stocks | n/a |
| Novel plant genotypes | n/a |
| Authentication | n/a |

