## [Peer Review file · Nature Medicine]

Predicting onset of symptomatic Alzheimer disease with plasma p-tau217 clocks

Corresponding Author: Dr Suzanne Schindler

Version 0:

Reviewer comments:

Reviewer #1

(Remarks to the Author)

This study evaluates the role of plasma p-tau217 in predicting onset of symptomatic Alzheimer's disease (AD) and the time from %p-tau217 positivity to onset of AD symptoms. They found utility in two models to predict these points of interest.

This is a very timely paper on an important topic in the field. Various clock models have been used to predict the onset of symptoms based on amyloid and tau PET, but the authors are now using %plasma p-tau217 as the predictor variable. Since plasma markers are more readily available, this has intrinsic interest. There are some methodological concerns.

From a clinical perspective, the study groups are somewhat difficult to follow. For example, in the last paragraph in the Introduction, they state, "in this study, we aim to use measurements from a single plasma sample to estimate not only the probability of a cognitively unimpaired individual developing symptomatic AD but also when they would be likely to develop symptoms." They are drawing a distinction between developing "symptomatic AD" and "developing symptoms." This is somewhat confusing to the reader especially when they are using two different participant populations. Do they mean symptom onset as measured by progression on the CDR from 0 to 0.5, or are they mentioning the actual historical onset of symptoms that the participants and families report? These data, I suspect, are collected quite differently in the two studies.

They also used the term "AD syndrome," but it is uncertain how this is defined. Do they mean that this is a clinical syndrome that the clinicians felt was likely due to AD, or is this the onset of a dementia pattern in people who are amyloid positive? Again, this is unclear in the manuscript and may vary considerably between the two sources of data.

In addition, they discuss individuals who develop cognitive impairment before %p-tau217 positivity. They concluded that these individuals had non-AD causes of cognitive impairment, but again, is this retrospective in individuals who did not fit the criteria for "AD syndrome," or were these individuals who just had cognitive impairment of some type and never developed a positive %p-tau217?

Later, they discuss people in the Knight ADRC sample with an AD or non-AD syndrome, but the details of this determination remain vague.

I recommend that the authors consider discussing continuous cognitive outcomes, such as AVLT t-scores or a global composite z-score, either as secondary outcomes or in sensitivity analyses. This would provide a more granular view of cognitive decline than diagnostic thresholds alone.

The authors use an alignment of amyloid and tau PET stages with %p-tau217. While they have proposed a reasonable staging scheme, it would be much more useful if they would engage the proposed Alzheimer's Association 2024 AD Staging Scheme which has biological stages, A, B, C and D. The proposed staging here is similar, but aligning it with other published work would be more useful for the reader.

The novelty of this paper lies more in its handling of clinical transitions than in the clock model itself, and I would encourage the authors to clarify how their approach differs from or improves upon non-linear mixed effects (NLME)-based trajectory modeling.

The study involves sophisticated statistical modeling. The authors have done a credible job in using the literature on clock stages as well as existing data. The challenge in this type of work pertains to determining when the "corner will be turned." That is, most of these biologic processes have a hockey stick-shaped progression, and the corner is the challenge. This paper is based on the premise that p-tau217 is measuring the same underlying process and will have the same behavior as, for example, amyloid PET. The key challenges are a) the threshold for the corner, b) the rate beyond the corner, and c) how

to interpret it. In this analysis, the methods for a and b above are difficult to comprehend. Statistically, many of the steps are completely ad hoc. It is difficult to endorse these steps because there are numerous assumptions involved.

The real question is whether the cutpoint and slopes that result work, and from the data, they appear to do reasonably well. The authors have chosen to encapsulate c above by giving each subject an estimated “age of transition” at which they have connected the rise in the curve and then used that as a covariate in ordinary survival models.

There may be two caveats with this approach. One has to be aware that this is only relevant to subjects with a p -tau above the given threshold and useless to all others. Due to measurement noise, it will be difficult for those who have only just crossed the threshold. Those who have two measurements over the threshold are most reliable. Their overall assessment of this process is a bit optimistic, and if their goal is to treat subjects very early in the disease, that is the precise range in the disease process where this method may be marginal. I think the conclusions are perhaps too optimistic given the data and the assumptions. While the approach is reasonable, it may have only limited applicability.

It is commendable that the authors have used two competing methods, the SILAC and the TIRA approach. They give somewhat different results, but they are close. That is comforting.

There are other methods that could be used and may or may not produce similar results. They have chosen to focus on a subset of individuals who qualify for the p -tau₂₁₇ measure. They do not account for dropouts or death.

In the Methods section, the authors note that variance in rate of change for p -tau₂₁₇ was assessed in participants with at least two values spaced one year apart. It would be helpful to clarify how many participants had three or more observations, as this affects the reliability of individual slope estimates. Additionally, please clarify whether the timing between observations was regular or varied significantly across individuals.

The authors state that “for models of age at symptom onset, individuals were included who were initially cognitively unimpaired but diagnosed with cognitive impairment due to AD at their last assessment and who developed symptoms after estimated plasma p -tau₂₁₇ positivity.” Please specify how many individuals met these criteria and were included in this analysis, as the sample size is critical for assessing the robustness of any inferences drawn.

While the filtering step on rate of change using GAM may help stabilize the rate function by excluding noisy or unstable portions of the data, it introduces potential concerns. First, this is a form of data-dependent subsetting, where the data are used twice: once to select the modeling range and again to fit the model within that range. This can induce selection bias, especially if the retained subset differs biologically or demographically from the full sample. As a result, the modeled rate function may no longer be representative of the full trajectory, and the resulting clock may not generalize well to individuals with p -tau₂₁₇ values outside the retained interval. Second, this strategy may lead to biased exclusion of the distributional extremes. For example, p -tau₂₁₇ values below 1% or above 10% may appear more variable due to lower density, but may still reflect meaningful early or late disease biology. Excluding these ends risks discarding informative data and could distort clock timing near clinically relevant thresholds. I suggest the authors explicitly acknowledge this limitation and, if feasible, provide a sensitivity analysis comparing models fit to the full range versus the trimmed range to assess the potential impact of this selection on clock estimates.

The manuscript would benefit from a more explicit discussion of co-pathology, particularly in relation to amyloid PET. This follows up on the earlier point about the AT staging framework and how tau biomarkers interact with amyloid burden. It would be helpful for the authors to clarify whether they considered the co-evolution of amyloid and tau pathology and how this may influence p -tau₂₁₇ dynamics or symptomatic conversion.

The statement that “for individuals with AD symptom onset after plasma p -tau₂₁₇ positivity, linear models predicted AD symptom onset as a function of the estimated age of p -tau₂₁₇ positivity” raises questions about model justification. Given the uncertainty in the estimated age of p -tau₂₁₇ positivity and potential variability across individuals, a linear relationship may be overly simplistic. The authors should justify the use of a linear model in this context and clarify whether any model diagnostics were conducted to assess its appropriateness.

In the subsection on modeling the probability of developing symptomatic AD, their use of left-, right-, and interval-censoring is statistically valid. However, the survival model treats the estimated age of p -tau₂₁₇ positivity—derived from either TIRA or SILA—as a fixed covariate, without accounting for the uncertainty in that estimate. This can lead to biased or overconfident hazard ratios. Furthermore, the primary analysis is restricted to individuals who were cognitively unimpaired at baseline and later developed AD, which induces survivor bias: individuals who were already symptomatic at enrollment are excluded, and those who died before developing symptoms (i.e., preclinical mortality) are not represented.

In summary, I think this is a very scholarly approach to a difficult problem. The authors have used two traditional databases, each of which has its own limitations. The clinical uncertainties mentioned above cloud some of the interpretations of the data. There are several assumptions that are made by the authors that are reasonable but do limit the applicability of the overall approach. The authors do state that this is a research effort and not for clinical use, which is appropriate. They also state that, due to instability in the data, it should be used for group data analysis purposes. However, in the Conclusion of the paper, they do state that models with this level of precision may assist in selecting participants for clinical trials targeting certain phases of preclinical disease. Again, this might be useful with certain assumptions but needs to be explicated when using the models.

The authors are to be applauded for an extremely academic approach to a real-world problem.

(Remarks on code availability)

Reviewer #3

(Remarks to the Author)

Summary: Petersen and colleagues used two previously established models (TIRA and SIRA) to model the age of p-tau_{217} and linked it to various outcomes, including age at onset of AD symptoms, clinical diagnosis, and cognitive impairment. Using two independent cohorts (ADNI and Knight ADRC), they have found relatively consistent results between the TIRA and SIRA models and concluded that the time until onset of AD symptoms can be estimated using a single blood test within a margin of error that is acceptable for use in clinical trials. The clinical value of such a clock model is very high and a great demonstration that plasma p-tau_{217} is a biomarker to predict disease progression of AD. I have the following questions and comments:

1: In Fig. 1, panels A and B are understandable since ADNI includes a higher proportion of cognitively impaired patients, resulting in a plateau effect where individuals with already elevated baseline values tend to show slower rates of change. However, panels C and D are somewhat counterintuitive, as the plateau effect appears more pronounced in the Knight ADRC dataset.

2: The paper frequently uses the term 'Predict,' including in the title. However, the analyses presented are primarily associative rather than predictive in the machine learning sense—that is, they do not involve modeling to forecast outcomes at the individual level. Notably, the SIRA and TIRA models utilize longitudinal data to estimate the age at positivity and then relate these estimates to other covariates within the same cohort (quantified using R and R^2), making it difficult to characterize these results as true predictions. I recommend that the authors consider adopting more precise terminology. A genuinely predictive model would require appropriate cross-validation or testing on independent datasets—for example, training the SIRA or TIRA model on the ADNI data and then evaluating its predictive performance on the Knight ADRC cohort.

3: This work builds upon prior research focused on the SIRA and TIRA models. I believe the authors should provide more detailed explanations of these models in the Introduction to help readers better understand and follow the results.

4: I think one comment at the high level, more conceptualization of the term: The manuscript frequently refers to the TIRA/SILA-derived trajectories as a “plasma p-tau_{217} clock.” The term clock is commonly used in the literature for broadly applicable biological age measures (e.g., epigenetic clocks, proteomic clocks) and may imply a global aging or organismal timing mechanism. In this study, the model represents a disease-specific staging tool for Alzheimer’s-related pathology, estimated from plasma p-tau_{217} trajectories. To avoid confusion, I suggest adopting terminology that more clearly conveys the scope and specificity of the measure, such as “plasma p-tau_{217} disease stage model,” “biomarker-based disease time axis,” or “ p-tau_{217} progression timeline.” If the authors wish to retain the term clock, it would be helpful to explicitly define it early in the manuscript as an Alzheimer’s disease-specific staging model to distinguish it from established biological aging clocks.

Minor points:

There are some typos in the paper. For example, Fig.1’s caption: “Clock models relating time to p-tau_{217} are shown for the Knight ADRC (C) and ADNI (D) cohorts and were created using two approaches: TIRA (green) and SILA (orange).” Please proofread it.

(Remarks on code availability)

The code is minimal and without detailed documentation. Therefore, it is hard to ensure reproducibility. Code for TIRA model is not presented; for SIRA, only MATLAB code is available, but the manuscript mentioned they used the R version.

Version 1:

Reviewer comments:

Reviewer #1

(Remarks to the Author)

The authors have provided an extensive response to the concerns raised in the earlier review. As such, I applaud them for their diligence and their attention to detail with regard to our comments. I have a few comments.

With all due respect to the authors, I must still disagree with their use of the term “AD syndrome.” They give several references as to how this term is used in the literature, but I think these references have been superseded. The term “AD syndrome” is nonspecific, and I believe the authors are referring to an amnesic presentation. However, as they know, an AD syndrome may include a visual variant, logopenic primary progressive aphasia, or a dysexecutive syndrome in addition to the amnesic presentation. While it is certainly true that the amnesic presentation is the most common clinical syndrome associated with AD, there are others, and as such, the term “AD syndrome” is nonspecific. It would be preferable for them to

describe the syndrome such as an amnesic syndrome if that is what they mean. The most recent version of the NACC diagnostic procedures describes clinical syndromes as outlined above coupled with potential biomarkers. The term "AD syndrome" is not used. As such, since they're using data from the Knight ADRC, they should use current terminology. Similarly, they describe symptomatic AD as the cognitive impairment with the AD syndrome in a context of a positive plasma biomarker. Again, they are referring to an amnesic syndrome with positive biomarkers. In addition, they state that the onset of AD symptoms is defined as the first clinical assessment with initially cognitively impaired individuals with positive AD biomarkers were found to be cognitively impaired (CDR > 0) with an AD syndrome. This really refers to their first diagnostic threshold rather than symptom onset. Symptom onset could occur many years prior to the first progression of CDR > 0, but that is not how they defined it. They have based all of their clinical judgments around the conversion of CDR > 0. In addition, when discussing individuals who did not have the AD syndrome, they state that individuals with a primary clinical diagnosis that did not include AD (such as Parkinson's disease, dementia or vascular dementia) were considered to have a non-AD syndrome. Again, it is unclear what constitutes Parkinson's disease dementia or vascular dementia. Are these clinical diagnoses, or are these clinical syndromes in the setting of extrapyramidal symptoms and/or vascular changes on MRI? The lack of precision of the clinical characterization is worrisome. I appreciate the authors considering the use of continuous measures to give a more granular appearance, and their use of the PACC is adequate.

Similarly, I appreciate the authors consideration of using the Alzheimer's Association 2024 AD Staging Scheme with Stages A, B, C and D. I think this will make the paper more consistent with other entities in the literature.

The authors have done a commendable job in addressing several of the statistical issues raised in the earlier review. In particular, it is imperative that they acknowledge that their analyses pertain to a restricted range of individuals with p-tau217. The discussion of the corners is extremely important, and they need to be certain that the reader appreciates that they are dealing with a relatively narrow range of p-tau217 measurements since the values at the extremes are less certain and of questionable validity. They acknowledge that the clock models are not very accurate at very low biomarker values, and this needs to be emphasized in the paper.

It is extremely important that the authors acknowledge that their analyses did not consider individuals who dropped out of the study or died. This is common in these types of models but also introduces an element of survival bias. This needs to be emphasized in the discussion.

The authors have acknowledged that their sample sizes for some of the modeling was quite limited. Again, this needs to be emphasized and is a cautionary note for interpreting these data.

The authors acknowledge that co-pathologies may complicate the interpretation of their model. While they are quite correct that p-tau217 reflects both amyloid and tau pathology, as the age of the participants increases, the likelihood of co-pathology complicating the interpretation of the models needs to be recognized.

In summary, the authors have done a credible job in responding to the earlier reviews. Some of the issues pertain to individual preferences for modeling and, as such, as long as they justify their approaches and clarify their modeling assumptions, this exercise is reasonable. The above concerns regarding clinical diagnostic terminology detract from the interpretability of the paper and will need to be clarified.

(Remarks on code availability)

I have not reviewed the code.

Reviewer #3

(Remarks to the Author)

The authors have responded to my previous comments. Congratulations on completing this important research!

(Remarks on code availability)

The code has been improved for potential use!

August 26, 2025

Reviewer #1:

This study evaluates the role of plasma p-tau217 in predicting onset of symptomatic Alzheimer's disease (AD) and the time from %p-tau217 positivity to onset of AD symptoms. They found utility in two models to predict these points of interest. This is a very timely paper on an important topic in the field. Various clock models have been used to predict the onset of symptoms based on amyloid and tau PET, but the authors are now using %plasma p-tau217 as the predictor variable. Since plasma markers are more readily available, this has intrinsic interest. There are some methodological concerns.

1. From a clinical perspective, the study groups are somewhat difficult to follow. For example, in the last paragraph in the Introduction, they state, "in this study, we aim to use measurements from a single plasma sample to estimate not only the probability of a cognitively unimpaired individual developing symptomatic AD but also when they would be likely to develop symptoms." They are drawing a distinction between developing "symptomatic AD" and "developing symptoms." This is somewhat confusing to the reader especially when they are using two different participant populations. Do they mean symptom onset as measured by progression on the CDR from 0 to 0.5, or are they mentioning the actual historical onset of symptoms that the participants and families report? These data, I suspect, are collected quite differently in the two studies.

Thank you for pointing out the need for further clarification of these terms. We have revised the manuscript throughout to establish clearer definitions, including in the results section:

"Symptomatic AD was defined to align with the clinical diagnosis of symptomatic AD: cognitive impairment (Clinical Dementia Rating [CDR]>0) with an AD syndrome (clinical features consistent with cognitive impairment caused by AD^{1,35-37}) in the context of biomarkers indicating the presence of AD pathology^{1,37}. The onset of AD symptoms was defined as the first clinical assessment when initially cognitively unimpaired (CDR=0) individuals with positive AD biomarkers (based on estimated %p-tau217) were found to be cognitively impaired (CDR>0) with an AD syndrome. Further, AD symptom onset was applied only to individuals who were cognitively impaired (CDR>0) with an AD syndrome at their last assessment, i.e., if an individual had transient cognitive impairment but returned to cognitively unimpaired or had a non-AD diagnosis at their last assessment, the earlier impairment was not considered to be the onset of AD symptoms." (page 7-8)

2. They also used the term "AD syndrome," but it is uncertain how this is defined. Do they mean that this is a clinical syndrome that the clinicians felt was likely due to AD, or is this the onset of a dementia pattern in people who are amyloid positive? Again, this is unclear in the manuscript and may vary considerably between the two sources of data.

We have further clarified that “AD syndrome” refers to the clinical features consistent with cognitive impairment caused by AD and provided references.

“Individuals with clinical features consistent with cognitive impairment caused by AD (e.g., insidious onset, slowly progressive decline, early amnesic impairment) were considered to have an AD syndrome^{1,35-37}. Individuals with a primary clinical diagnosis that did not include AD (such as Parkinson disease dementia and vascular dementia) were considered to have a non-AD syndrome. The assessment of clinical syndrome was made by clinicians who were blinded to biomarker results and determinations were based solely on clinical presentation and established diagnostic criteria, and was recorded as the primary clinical diagnosis^{1,35-37}.” (page 16)

3. In addition, they discuss individuals who develop cognitive impairment before %p-tau217 positivity. They concluded that these individuals had non-AD causes of cognitive impairment, but again, is this retrospective in individuals who did not fit the criteria for “AD syndrome,” or were these individuals who just had cognitive impairment of some type and never developed a positive %p-tau217?

We made these distinctions to examine cognitively impaired individuals who would be clinically diagnosed with symptomatic AD (AD syndrome and positive AD biomarkers) and contrast this with individuals who would not receive a clinical diagnosis of AD (e.g., non-AD syndrome and/or negative AD biomarkers). This is a framework based on established AD definitions of AD. Of course, cognitive impairment is complex and individuals may have cognitive impairment from numerous etiologies. We discuss clinical heterogeneity as a limitation of our approach:

“Symptomatic AD was defined as cognitive impairment with an AD syndrome in the context of an estimated positive %p-tau217 value. The threshold for %p-tau217 positivity corresponds to an amyloid PET Centiloid value of 20, below which very few individuals have symptoms due to AD^{25,47}, making it unlikely that individuals with an estimated negative %p-tau217 value have cognitive impairment due to AD pathology. However, occasional individuals may have discrepant %p-tau217 values. Notably, results were also shown for individuals with an AD syndrome with an estimated negative %p-tau217 value and those with non-AD dementia syndromes.” (page 14)

Later, they discuss people in the Knight ADRC sample with an AD or non-AD syndrome, but the details of this determination remain vague.

We have further clarified that “AD syndrome” refers to the clinical features consistent with cognitive impairment caused by AD and provided references. Individuals with a primary clinical diagnosis other than AD were considered to have a non-AD syndrome.

“Individuals with clinical features consistent with cognitive impairment caused by AD (e.g., insidious onset, slowly progressive decline, early amnesic impairment) were considered to have an AD syndrome^{1,35-37}. Individuals with a primary clinical diagnosis that did not include AD (such as Parkinson disease dementia and vascular dementia) were

considered to have a non-AD syndrome. The assessment of clinical syndrome was made by clinicians who were blinded to biomarker results and determinations were based solely on clinical presentation and established diagnostic criteria, and was recorded as the primary clinical diagnosis^{1,35-37}.” (page 16)

4. I recommend that the authors consider discussing continuous cognitive outcomes, such as AVLT t-scores or a global composite z-score, either as secondary outcomes or in sensitivity analyses. This would provide a more granular view of cognitive decline than diagnostic thresholds alone.

We appreciate the suggestion to discuss continuous cognitive outcomes as an alternative outcome to the binary outcomes we examined. We have now added to the Discussion regarding this point.

“Future investigations could also explore continuous cognitive measures that identify subtle cognitive changes that occur before the threshold for clinical diagnosis.”
(page 12)

Additionally, we performed a sensitivity analysis using the Preclinical Alzheimer Cognitive Composite, a continuous outcome. This analysis was performed in the ADNI cohort using Preclinical Alzheimer’s Cognitive Composite (PACC; Digit Symbol Substitution version), a continuous measure. We defined PACC>-1 as cognitively normal subset of individuals. We then calculated the z-score of this subset and looking at the change in PACC from z-score>-1 to ≤-1. Survival analyses demonstrated that individuals who became %p-tau217 positive at older ages had a shorter time to impairment on the PACC, similar to the primary analysis findings in this manuscript (see figure below).

We also examined associations between estimated age at plasma %p-tau217 positivity and age at impairment on the PACC. Among individuals who developed impairment on the PACC, moderate associations were observed between conversion age and estimated age at %p-tau217 positivity using both TIRA and SILA models (see figure below). These analyses provide confidence in the robustness of our findings, although given that these analyses could be performed many different ways and we hope to perform similar analyses in future analyses, we chose not to include them in the current manuscript.

5. The authors use an alignment of amyloid and tau PET stages with %p-tau217. While they have proposed a reasonable staging scheme, it would be much more useful if they would engage the proposed Alzheimer's Association 2024 AD Staging Scheme which has biological stages, A, B, C and D. The proposed staging here is similar, but aligning it with other published work would be more useful for the reader.

We appreciate the suggestion to align our plasma %p-tau217 clock models to the new Alzheimer's Association 2024 AD Staging Scheme and have now implemented this in our revised manuscript (see the revised **Extended Figure 4** below). We replaced our previous A-T_{early}-T_{late}- notation with the standardized Stage A (normal biomarkers), Stage B (AD pathologic change), Stage C (AD), and Stage D (advanced AD) terminology. The analysis presented in the attached figure has been reanalyzed and replotted using these biological stages, with corresponding updates made throughout the Results section to demonstrate how plasma %p-tau217 levels and, importantly, clock estimates progress across stages (A to D). We have also updated our Methods section to reflect the new terminology.

6. The novelty of this paper lies more in its handling of clinical transitions than in the clock model itself, and I would encourage the authors to clarify how their approach differs from or improves upon non-linear mixed effects (NLME)-based trajectory modeling.

We have added discussion of the utility of clock models relative to non-linear mixed effects models:

“While statistical models typically include age as a covariate, the relationship between plasma %p-tau217 levels and symptom onset is complex and may not be well captured by linear or non-linear models, although age-stratified analyses may be helpful. Non-linear mixed effects models characterize population-level trajectories with individual random effects, but clock models explicitly convert biomarker levels into individualized estimates that are intuitive (e.g., years since biomarker positivity) and may reveal important findings such as the marked effect of age at plasma %p-tau217 positivity on the age at AD symptom onset.” (pages 12-13)

7. The study involves sophisticated statistical modeling. The authors have done a credible job in using the literature on clock stages as well as existing data. The challenge in this type of work pertains to determining when the “corner will be turned.” That is, most of these

biologic processes have a hockey stick-shaped progression, and the corner is the challenge. This paper is based on the premise that p-tau217 is measuring the same underlying process and will have the same behavior as, for example, amyloid PET. The key challenges are a) the threshold for the corner, b) the rate beyond the corner, and c) how to interpret it. In this analysis, the methods for a and b above are difficult to comprehend. Statistically, many of the steps are completely ad hoc. It is difficult to endorse these steps because there are numerous assumptions involved.

We have centered the models for this paper on at the age at biomarker positivity aligned with a relatively widely used standard (amyloid PET Centiloid 20), which is beyond the “corner,” rather than an approach that centered them at the “corner,” or “tipping point,” that was used in a previous paper (Schindler *et al. Neurology* 2021). We have added **Extended Figure 1**, which shows the “corners,” to demonstrate that the %p-tau217 and p-tau217 rates of change are relatively consistent within a certain window and consistent. Notably, we also demonstrated that our approach is not limited to the specific %p-tau217 or p-tau217 measures by successfully implementing similar clock models using the Fujirebio Lumipulse p-tau217/A β 42 measure. The p-tau217/A β 42 measure showed moderate associations with symptom onset confirming that our clock modeling.

Assumptions are required for all mathematical models. A major strength of our manuscript was employing two different clock model approaches, SILA and TIRA, which have different underlying mathematical assumptions. Despite all the complexities, the clock models performed similarly, providing reassurance that the underlying patterns are real. Furthermore, multiple approaches to examine the relationship between estimated age at %p-tau217 positivity and AD symptom onset yielded similar findings, again providing reassurance that these approaches, while complex, are finding real patterns.

8. The real question is whether the cutpoint and slopes that result work, and from the data, they appear to do reasonably well. The authors have chosen to encapsulate c above by giving each subject an estimated “age of transition” at which they have connected the rise in the curve and then used that as a covariate in ordinary survival models. There may be two caveats with this approach. One has to be aware that this is only relevant to subjects with a p -tau above the given threshold and useless to all others. Due to measurement noise, it will be difficult for those who have only just crossed the threshold. Those who have two measurements over the threshold are most reliable. Their overall assessment of this process is a bit optimistic, and if their goal is to treat subjects very early in the disease, that is the precise range in the disease process where this method may be marginal. I think the conclusions are perhaps too optimistic given the data and the assumptions. While the approach is reasonable, it may have only limited applicability.

We agree that the clock models are not accurate at very low biomarker values—this is why we have restricted the clock model range to the values over which there is a consistent rate of change (see response to comment 7 and Extended Figure 1). However, these models are showing good performance over much of the range of p -tau217 values that are found in preclinical AD, including even values several years before p -tau217 positivity.

9. It is commendable that the authors have used two competing methods, the SILAC and the TIRA approach. They give somewhat different results, but they are close. That is comforting. There are other methods that could be used and may or may not produce similar results. They have chosen to focus on a subset of individuals who qualify for the p -tau217 measure. They do not account for dropouts or death.

We thank the reviewer for this thoughtful comment. We agree that the consistency between the two modeling approaches (TIRA and SILA) is reassuring and supports the robustness of clock models. Work with clock models based on longitudinal amyloid PET data has also found relatively consistency across different mathematical approaches (Betthausen *et al. Brain* 2022). Regarding participant selection and attrition, we have added text to the limitations section to address these points:

“Additional limitations include that individuals included in the study had a variety of clinical diagnoses that were grouped together for analyses and the models do not reflect the full complexity of clinical symptoms. Participants in the study largely identified as non-Hispanic White, which may limit the generalizability of these models to other groups, especially groups with different rates of non-AD co-pathologies. Further, like other longitudinal aging studies, our analysis did not explicitly model participant dropout or death, which could introduce survival bias if individuals who develop more rapid cognitive decline are more likely to discontinue participation.” (Page 14)

10. In the Methods section, the authors note that variance in rate of change for p -tau217

was assessed in participants with at least two values spaced one year apart. It would be helpful to clarify how many participants had three or more observations, as this affects the reliability of individual slope estimates. Additionally, please clarify whether the timing between observations was regular or varied significantly across individuals.

We have added these details to the results section:

“Knight ADRC participants had a median of 7.1 years (IQR 5.0-11.0 years) from the first to last plasma collection. Of these participants, 310 of 506 (61.3%) provided three or more plasma samples. ADNI participants had a median of 5.0 years (IQR 4.0-6.5 years) from the first to last plasma collection. Of these participants, 404 of 406 (99.5%) provided three or more plasma samples.” (page 6)

11. The authors state that “for models of age at symptom onset, individuals were included who were initially cognitively unimpaired but diagnosed with cognitive impairment due to AD at their last assessment and who developed symptoms after estimated plasma %p-tau217 positivity.” Please specify how many individuals met these criteria and were included in this analysis, as the sample size is critical for assessing the robustness of any inferences drawn.

These sizes are reported in the manuscript text, **Supplementary Table 2**, and are also shown in the **Figure 4**.

“For the Knight ADRC cohort, models included 59 individuals using TIRA and 61 individuals using SILA clock models; for the ADNI cohort, models included 20 individuals using TIRA and 22 individuals using SILA (see **Supplementary Table 2** for cohort characteristics).” (page 9)

12. While the filtering step on rate of change using GAM may help stabilize the rate function by excluding noisy or unstable portions of the data, it introduces potential concerns. First, this is a form of data-dependent subsetting, where the data are used twice: once to select the modeling range and again to fit the model within that range. This can induce selection bias, especially if the retained subset differs biologically or demographically from the full sample. As a result, the modeled rate function may no longer be representative of the full trajectory, and the resulting clock may not generalize well to individuals with %p-tau217 values outside the retained interval. Second, this strategy may lead to biased exclusion of the distributional extremes. For example, %p-tau217 values below 1% or above 10% may appear more variable due to lower density, but may still reflect meaningful early or late disease biology. Excluding these ends risks discarding informative data and could distort clock timing near clinically relevant thresholds. I suggest the authors explicitly acknowledge this limitation and, if feasible, provide a sensitivity analysis comparing models fit to the full range versus the trimmed range to assess the potential impact of this selection on clock estimates.

The reviewer notes that the modeled function rate may not be representative of the full trajectory,

and the resulting clock may not generalize well to individuals with %p-tau217 values outside the retained interval. This is why we have not calculated age at %p-tau217 positivity for individuals with values outside of the restricted range. It is true that excluding the ends may discard informative data, but the high variance at the low and high end (as well as sparse data at the high end) would significantly reduce the uncertainty of estimates. The range represents much of preclinical AD and thus we think it is still informative.

We have now added a figure showing clock models constructed with all longitudinal data that are directly compared to models made with restricted range of values.

“Similar clock models were constructed using all longitudinal data (**Supplementary Figure 1**). At very low %p-tau217 values (<1.06%) there were highly unstable TIRA time estimates in the ADNI cohort. Longitudinal data from individuals with high %p-tau217 values was sparse (**Figure 1A-B**), providing less certain clock estimates for higher values. Further, there were rapid increases in %p-tau217 at high values (>10.45%) that could make time estimates unstable. These features provided further rationale to use clocks constructed with the restricted range of %p-tau217 values (1.06-10.45%).” (page 6-7)

13: The manuscript would benefit from a more explicit discussion of co-pathology, particularly in relation to amyloid PET. This follows up on the earlier point about the AT staging framework and how tau biomarkers interact with amyloid burden. It would be helpful for the authors to clarify whether they considered the co-evolution of amyloid and tau pathology and how this may influence %p-tau217 dynamics or symptomatic conversion.

We have added clarifying statements in this manuscript explaining that our clock models are built on a single biomarker, %p-tau217, which reflects combined pathological processes. While our models do not explicitly model the joint dynamics of amyloid and tau, the strong associations of %p-tau217 with both PET measures suggest these models capture an integrated view of disease progression. We also highlight that %p-tau217 dynamics likely represent the intertwined progression of these pathologies, and that additional age-related co-pathologies, like cerebrovascular disease, may influence clinical outcomes, especially the shorter time to symptom onset in older individuals. Our analysis of AT stage alignment further supports that %p-tau217 tracks the co-evolution of amyloid and tau pathology across the AD continuum. These clarifications enhance our understanding of how the approach in this work relates to the complex relationship of AD pathologies.

“Although our clock models use the single biomarker %p-tau217, its strong associations with amyloid and tau PET effectively integrate the pathological processes of amyloid plaques and neurofibrillary tangles into the models. Importantly, %p-tau217 dynamics likely capture the intertwined progression of both amyloid and tau pathology. Furthermore, the shorter interval from %p-tau217 positivity to symptom onset observed in older individuals may, in part, reflect the influence of age-related co-pathologies such as cerebrovascular disease and other neurodegenerative diseases⁴⁶. Recognizing the impact of these additional pathologies is crucial, as they may modify clinical trajectories beyond the core AD pathology. Future work incorporating complementary biomarkers of amyloid, tau, and other pathologies will be important for improving the accuracy and applicability of these models.” (pages 13)

14: The statement that “for individuals with AD symptom onset after plasma %p-tau217 positivity, linear models predicted AD symptom onset as a function of the estimated age of %p-tau217 positivity” raises questions about model justification. Given the uncertainty in the estimated age of %p-tau217 positivity and potential variability across individuals, a linear relationship may be overly simplistic. The authors should justify the use of a linear model in this context and clarify whether any model diagnostics were conducted to assess its appropriateness.

Model diagnostics were conducted to justify the linear modeling approach. We evaluated key regression assumptions through Shapiro-Wilk tests for residual normality and Breusch-Pagan tests for homoscedasticity, both of which were satisfied across all cohort-method combinations. To address concerns about over-simplification, we explicitly tested for non-linear relationships by comparing linear models against quadratic and cubic polynomial alternatives using AIC-based model selection and F-tests. Results consistently favored the linear specification, with no significant improvement from higher-order polynomial terms (all F-test p-values > 0.175). Additionally, influence diagnostics including Cook's distance confirmed model robustness to potential outliers. These comprehensive diagnostics, detailed in **Supplementary Table 4**, provide empirical support for the linear modeling approach.

15. In the subsection on modeling the probability of developing symptomatic AD, their use of left-, right-, and interval-censoring is statistically valid. However, the survival model treats the estimated age of %p-tau217 positivity—derived from either TIRA or SILA—as a fixed covariate, without accounting for the uncertainty in that estimate. This can lead to biased or overconfident hazard ratios. Furthermore, the primary analysis is restricted to individuals who were cognitively unimpaired at baseline and later developed AD, which induces survivor bias: individuals who were already symptomatic at enrollment are excluded, and those who died before developing symptoms (i.e., preclinical mortality) are not represented.

We have noted in our Methods that "survival models treated estimated age at %p-tau217 positivity as a fixed covariate from the clock model estimation." However, the survivor bias concern is addressed by our sensitivity analysis presented in **Supplementary Figure 4**, which we have now emphasized more clearly in the Results by emphasizing that the analyses "confirmed these associations with better discrimination and addressed potential survivor bias." This analysis includes all participants regardless of baseline cognitive status using left-censoring for those with cognitive impairment at enrollment, and importantly associations remain robust, and discrimination actually improves (C-indices of 0.821-0.828 for Knight ADRC and 0.672-0.707 for ADNI compared to 0.784-0.790 and 0.730-0.750 in the primary analysis). While we cannot account for preclinical mortality that may occur before study enrollment in observational cohorts, our left-censored analysis does mitigate survivor bias by including participants across the cognitive spectrum. Importantly, we see improved discrimination in this more inclusive analysis which actually strengthens our confidence in our findings.

16. In summary, I think this is a very scholarly approach to a difficult problem. The authors have used two traditional databases, each of which has its own limitations. The clinical uncertainties mentioned above cloud some of the interpretations of the data. There are several assumptions that are made by the authors that are reasonable but do limit the applicability of the overall approach. The authors do state that this is a research effort and not for clinical use, which is appropriate. They also state that, due to instability in the data, it should be used for group data analysis purposes. However, in the Conclusion of the paper, they do state that models with this level of precision may assist in selecting participants for clinical trials targeting certain phases of preclinical disease. Again, this might be useful with certain assumptions but needs to be explicated when using the models.

The authors are to be applauded for an extremely academic approach to a real-world problem.

We thank this reviewer for their detailed review of our manuscript and appreciation of the difficulty of this problem and the effort that went into building and interpreting these models.

Reviewer #3 (Remarks to the Author):

Summary: Petersen and colleagues used two previously established models (TIRA and SIRA) to model the age of %p-tau217 and linked it to various outcomes, including age at onset of AD symptoms, clinical diagnosis, and cognitive impairment. Using two independent cohorts (ADNI and Knight ADRC), they have found relatively consistent results between the TIRA and SIRA models and concluded that the time until onset of AD symptoms can be estimated using a single blood test within a margin of error that is acceptable for use in clinical trials. The clinical value of such a clock model is very high and a great demonstration that plasma %p-tau217 is a biomarker to predict disease progression of AD. I have the following questions and comments:

1. In Fig. 1, panels A and B are understandable since ADNI includes a higher proportion of cognitively impaired patients, resulting in a plateau effect where individuals with already elevated baseline values tend to show slower rates of change. However, panels C and D are somewhat counterintuitive, as the plateau effect appears more pronounced in the Knight ADRC dataset.

The reviewer notes an apparent discordance between the slope in the rate of change in panels A-B versus the resulting clock models in panels C-D. However, transformation of these rates into cumulative time curves can amplify subtle dataset differences that influence the final time curve shapes. This discordance illustrates how biomarker clock models capture different aspects of disease progression beyond simple rate patterns.

2. The paper frequently uses the term 'Predict,' including in the title. However, the analyses presented are primarily associative rather than predictive in the machine learning sense—that is, they do not involve modeling to forecast outcomes at the individual level. Notably, the SIRA and TIRA models utilize longitudinal data to estimate the age at positivity and then relate these estimates to other covariates within the same cohort (quantified using R and R²), making it difficult to characterize these results as true predictions. I recommend that the authors consider adopting more precise terminology. A genuinely predictive model would require appropriate cross-validation or testing on independent datasets—for example, training the SIRA or TIRA model on the ADNI data and then evaluating its predictive performance on the Knight ADRC cohort.

We did conduct the type of cross-validation that the reviewer mentions. As detailed in **Extended Table 3**, we trained models in the Knight ADRC cohort and tested them on the ADNI cohort, yielding moderate associations with adjusted R² of 0.467-0.463 and median average errors of 3.0-3.2 years. Additionally, we also performed the reverse cross-validation, training models in ADNI and testing on Knight ADRC, showing moderate associations with adjusted R² of 0.509-0.577 and median average errors of 3.6-3.7 years.

We agree that more precise language would strengthen the manuscript and we have refined our use of "prediction" where appropriate. In particular, we modified our language when distinguishing between our external validation across independent cohorts and our association modeling within cohorts using longitudinal data.

3. This work builds upon prior research focused on the SIRA and TIRA models. I believe the authors should provide more detailed explanations of these models in the Introduction to help readers better understand and follow the results.

We thank the reviewer for this suggestion and we have updated the Introduction to provide more conceptual background on biomarker clock models, including a clear definition that distinguishes these models from general biological aging clocks and categorical staging based on multiple biomarkers.

“Interestingly, once amyloid plaques and neurofibrillary tangles start to accumulate, the burden of pathology follows a remarkably consistent trajectory across individuals^{2,10-13}. These consistent trajectories enable the creation of clock models that relate levels of amyloid or tau PET signal to time and allow for estimation of when individuals developed amyloid or tau PET abnormality^{2,11-15}. Unlike general biological aging clocks or categorical staging based on multiple biomarkers, clock models track disease progression with a specific biomarker, thereby providing granular and intuitive time-based staging^{1,10,16,17}. Clock models allow alignment of trajectories to a reference point (e.g., amyloid or tau PET positivity) that reduces heterogeneity compared to models using chronological age alone^{13,16}. Further, the age at amyloid or tau abnormality or positivity estimated by clock models can be used to estimate the age at AD symptom onset^{10,12,13,15,18,19}. Knowing not just if but when AD symptoms are likely to manifest would be very useful to clinical trials that aim to prevent or slow progression to symptomatic AD²⁰.” (page 4)

Further, we named the clock model approaches in the Introduction (they are described in more detail in the methods):

“Two approaches were used to create %p-tau217 clock models: Temporal Integration of Rate Accumulation (TIRA)^{10,13}, which integrates the inverse of the modeled rate of change, and Sampled Iterative Local Approximation (SILA)^{12,18}, which uses discrete rate sampling and Euler's method for numerical integration.” (page 5)

4. I think one comment at the high level, more conceptualization of the term: The manuscript frequently refers to the TIRA/SILA-derived trajectories as a “plasma %p-tau217 clock.” The term clock is commonly used in the literature for broadly applicable biological age measures (e.g., epigenetic clocks, proteomic clocks) and may imply a global aging or organismal timing mechanism. In this study, the model represents a disease-specific staging tool for Alzheimer’s-related pathology, estimated from plasma %p-tau217

trajectories. To avoid confusion, I suggest adopting terminology that more clearly conveys the scope and specificity of the measure, such as “plasma %p-tau217 disease stage model,” “biomarker-based disease time axis,” or “p-tau217 progression timeline.” If the authors wish to retain the term clock, it would be helpful to explicitly define it early in the manuscript as an Alzheimer’s disease–specific staging model to distinguish it from established biological aging clocks.

We appreciate this thoughtful comment about terminology. We have now described different types of clocks in the introduction and more clearly introduced the %p-tau217 clock term (see the response to the previous comment).

5. There are some typos in the paper. For example, Fig.1’s caption: “Clock models relating time to %p-tau217 are shown for the Knight ADRC (C) and ADNI (D) cohorts and were created using two approaches: TIRA (green) and SILA (orange).” Please proofread it.

We have identified and corrected this grammatical error and have conducted a comprehensive proofreading of the entire manuscript to identify and fix additional typographical and grammatical errors throughout the text, figures, and supplementary materials. We appreciate the reviewer’s attention to detail.

6. The code is minimal and without detailed documentation. Therefore, it is hard to ensure reproducibility. Code for TIRA model is not presented; for SIRA, only MATLAB code is available, but the manuscript mentioned they used the R version.

The associated code, including for the TIRA model, has been deposited on Github and we have added much more thorough annotation of the code:

https://github.com/WashUFluidBiomarkers/plasma_ptau217_time

The MATLAB version of the SILA code from the Betthausser lab is here:

<https://github.com/Betthausser-Neuro-Lab/SILA-AD-Biomarker>

The R version of the SILA code only recently became available (at the link below), but it was not used in this study: <https://github.com/bilgelm/silaR>

We think our revised manuscript is responsive to the reviewers’ concerns and now much stronger with the additions they recommended. We are optimistic that it will be a highly informative and impactful publication, and we hope that the Editors will find it to be acceptable for publication in *Nature Medicine*.

Sincerely,

Suzanne E. Schindler, MD, PhD

October 29, 2025

Reviewer #1:

The authors have provided an extensive response to the concerns raised in the earlier review. As such, I applaud them for their diligence and their attention to detail with regard to our comments. I have a few comments.

With all due respect to the authors, I must still disagree with their use of the term “AD syndrome.” They give several references as to how this term is used in the literature, but I think these references have been superseded. The term “AD syndrome” is nonspecific, and I believe the authors are referring to an amnesic presentation. However, as they know, an AD syndrome may include a visual variant, logopenic primary progressive aphasia, or a dysexecutive syndrome in addition to the amnesic presentation. While it is certainly true that the amnesic presentation is the most common clinical syndrome associated with AD, there are others, and as such, the term “AD syndrome” is nonspecific. It would be preferable for them to describe the syndrome such as an amnesic syndrome if that is what they mean. The most recent version of the NACC diagnostic procedures describes clinical syndromes as outlined above coupled with potential biomarkers. The term “AD syndrome” is not used. As such, since they’re using data from the Knight ADRC, they should use current terminology.

Similarly, they describe symptomatic AD as the cognitive impairment with the AD syndrome in a context of a positive plasma biomarker. Again, they are referring to an amnesic syndrome with positive biomarkers. In addition, they state that the onset of AD symptoms is defined as the first clinical assessment with initially cognitively impaired individuals with positive AD biomarkers were found to be cognitively impaired ($CDR > 0$) with an AD syndrome. This really refers to their first diagnostic threshold rather than symptom onset. Symptom onset could occur many years prior to the first progression of $CDR > 0$, but that is not how they defined it. They have based all of their clinical judgments around the conversion of $CDR > 0$.

In addition, when discussing individuals who did not have the AD syndrome, they state that individuals with a primary clinical diagnosis that did not include AD (such as Parkinson’s disease, dementia or vascular dementia) were considered to have a non-AD syndrome. Again, it is unclear what constitutes Parkinson’s disease dementia or vascular dementia. Are these clinical diagnoses, or are these clinical syndromes in the setting of extrapyramidal symptoms and/or vascular changes on MRI? The lack of precision of the clinical characterization is worrisome.

We appreciate the reviewer's concern about clinical terminology and diagnostic precision. To address this, we have added Supplementary Tables 8 and 9, which provide more details on the primary clinical diagnoses in both the Knight ADRC and ADNI cohorts. Notably, the clinical diagnosis at the Knight ADRC (coded under the variable DX1) refers to a clinical syndrome judged by the clinician to be consistent with an etiology of AD, which includes not only an amnesic presentation but also logopenic aphasia, posterior cortical dysfunction, or dysexecutive presentations. So, we are referring to multiple presentations consistent with AD, although these specific presentations are not always clearly defined in our dataset. Furthermore, we added more details about the suspected non-AD syndromes in the tables, which included multiple categories such as “uncertain etiology” and “Dementia with Lewy Bodies” or

“Vascular dementia.” These clinical diagnoses are heterogeneous, making precise clinical characterization of hundreds of assessments over time impossible. In ADNI, the clinician diagnosis is lumped into even larger categories of MCI due to AD (or not) and dementia due to AD (or not). While it may be unsatisfying, interpreting this data requires lumping the data into groups (AD or non-AD) to provide statistical power. The underlying heterogeneity is reflected in the error of estimates. We have clarified this in the results and methods, “Symptomatic AD was defined to align with the clinical diagnosis of symptomatic AD: cognitive impairment (Clinical Dementia Rating [CDR]>0) with an AD syndrome (clinical features consistent with cognitive impairment caused by AD including amnesic, logopenic aphasia, posterior cortical dysfunction, or dysexecutive presentations^{1,35-37}) in the context of biomarkers indicating the presence of AD pathology^{1,37}.” (page 7)

The reviewer correctly notes that symptom onset could occur many years before the first recorded CDR>0, although inspection of the longitudinal data (as in Figure 5) demonstrates that most individuals underwent annual clinical assessment and therefore it is unlikely that symptom onset occurred many years before cognitive impairment in most individuals. Further, to account for variable time between clinical assessments, the time of AD symptom onset was interval-censored between the last cognitively unimpaired assessment and the first symptomatic AD assessment.

I appreciate the authors considering the use of continuous measures to give a more granular appearance, and their use of the PACC is adequate.

Thank you for your positive feedback regarding our use of continuous measures and the PACC. We appreciate your acknowledgment and are glad that our approach is considered adequate.

Similarly, I appreciate the authors consideration of using the Alzheimer’s Association 2024 AD Staging Scheme with Stages A, B, C and D. I think this will make the paper more consistent with other entities in the literature.

Thank you for your positive feedback on our consideration and use of the Alzheimer’s Association 2024 AD Staging Scheme. We agree that aligning with this framework enhances the consistency of our work with the current literature.

The authors have done a commendable job in addressing several of the statistical issues raised in the earlier review.

Thank you for your kind words regarding our efforts to address the statistical issues raised in the previous review. We appreciate your acknowledgment and are glad our revisions have satisfactorily resolved these concerns.

In particular, it is imperative that they acknowledge that their analyses pertain to a restricted range of individuals with p-tau217. The discussion of the corners is extremely important, and they need to be certain that the reader appreciates that they are dealing with a relatively narrow range of p-tau217 measurements since the values at the extremes are less certain and of questionable validity. They acknowledge that the clock models are not very accurate at very low biomarker values, and this needs to be emphasized in the paper.

Thank you for highlighting this important point. In response, we have revised the Discussion to explicitly acknowledge that our analyses did not account for individuals who dropped out or died during the study

period. We now discuss the potential for survival bias introduced by this limitation, which is common in these types of models, and emphasize its implications for the interpretation of our findings.

It is extremely important that the authors acknowledge that their analyses did not consider individuals who dropped out of the study or died. This is common in these types of models but also introduces an element of survival bias. This needs to be emphasized in the discussion.

Thank you for highlighting the need to discuss the impact of participant dropout and death on survival bias. We have expanded the limitations section in the Discussion to specifically emphasize that our analyses did not model dropout or death, and we now address the implications of survival bias resulting from this limitation. We note that this bias may lead to underestimation of decline among the most vulnerable individuals.

“Further, like other longitudinal aging studies, our analysis did not explicitly model participant dropout or death, which could introduce survival bias if individuals who develop more rapid cognitive decline are more likely to discontinue participation. This potential for survival bias is an important consideration when interpreting our results, as it may lead to underestimation of decline among the most vulnerable individuals.” (page 14)

The authors have acknowledged that their sample sizes for some of the modeling was quite limited. Again, this needs to be emphasized and is a cautionary note for interpreting these data. The authors acknowledge that co-pathologies may complicate the interpretation of their model.

Thank you for your comment highlighting the need to further emphasize the impact of limited sample sizes and co-pathologies on the interpretation of our results. In response, we have updated the limitations section to clarify that interpretation of estimates for smaller subgroups or those with mixed clinical presentations may be affected by limited sample sizes and co-pathologies. This additional note serves as a caution for readers and addresses your concern regarding the reliability of model performance in these groups. “Interpretation of estimates for smaller subgroups or those with mixed clinical presentations may be affected by limited sample sizes and co-pathologies.” (page 14)

While they are quite correct that p-tau217 reflects both amyloid and tau pathology, as the age of the participants increases, the likelihood of co-pathology complicating the interpretation of the models needs to be recognized.

Thank you for highlighting the importance of recognizing age-related increases in co-pathology and their impact on model interpretation. In response, we have added a statement to the Discussion noting that co-pathologies are more common in older individuals and may further complicate the interpretation of p-tau217 levels with increasing age. This addition aims to clarify for readers how age-related factors may influence biomarker interpretation:

“Importantly, we found that older individuals have a markedly shorter time until AD symptom onset after developing plasma %p-tau217 positivity. This is consistent with our previous work predicting symptom onset with amyloid PET¹⁰, where we found that older individuals developed symptoms at a lower amyloid PET burden. Age-related brain changes, including age-related increases in the prevalence of co-pathologies that affect the relationships between clinical symptoms and AD pathology⁴⁶, may underly this effect. As age increases, co-pathologies become more common and may further complicate the

interpretation of %p-tau217 levels in older individuals. This finding has significant implications for clinical trials: individuals with the same plasma %p-tau217 values likely have very different risks of developing cognitive impairment over a 3-5 year period depending on their age.” (page 12-13)

In summary, the authors have done a credible job in responding to the earlier reviews. Some of the issues pertain to individual preferences for modeling and, as such, as long as they justify their approaches and clarify their modeling assumptions, this exercise is reasonable. The above concerns regarding clinical diagnostic terminology detract from the interpretability of the paper and will need to be clarified.

Thank you for your very thoughtful and helpful review!

I have not reviewed the code.

Thank you for your note. We acknowledge that the code was not reviewed and appreciate your other feedback on the manuscript.

Reviewer #2:

The authors have responded to my previous comments. Congratulations on completing this important research!

Thank you for your positive feedback and for recognizing our efforts in addressing your previous comments. We appreciate your thoughtful review and encouragement.

The code has been improved for potential use!

Thank you for noting the improvements to the code. We appreciate your recognition and are pleased that the code now meets the standards for potential use.

We think our revised manuscript is responsive to the reviewers' concerns and now much stronger with the additions they recommended. We are optimistic that it will be a highly informative and impactful publication, and we hope that the Editors will find it to be acceptable for publication in Nature Medicine.

Sincerely,

Suzanne E. Schindler, MD, PhD